# Transformers Can Navigate Mazes With Multi-Step Prediction

## Abstract

Despite their remarkable success in language modeling, transformers trained to predict the next token in a sequence struggle with long-term planning. This limitation is particularly evident in tasks requiring foresight to plan multiple steps ahead such as maze navigation. The standard next *single* token prediction objective, however, offers no explicit mechanism to predict multiple steps ahead—or revisit the path taken so far. Consequently, in this work we study whether explicitly predicting multiple steps ahead (and backwards) can improve transformers' maze navigation. We train parameter-matched transformers from scratch, under identical settings, to navigate mazes of varying types and sizes with standard next token prediction and MLM-$\mathcal{U}$, an objective explicitly predicting multiple steps ahead and backwards. We find that MLM-$\mathcal{U}$ considerably improves transformers' ability to navigate mazes compared to standard next token prediction across maze types and complexities. We also find MLM-$\mathcal{U}$ training is $4\times$ more sample efficient and converges $2\times$ faster in terms of GPU training hours relative to next token training. Finally, for more complex mazes we find MLM-$\mathcal{U}$ benefits from scaling to larger transformers. Remarkably, we find transformers trained with MLM-$\mathcal{U}$ outperform larger transformers trained with next token prediction using additional supervision from A* search traces. We hope these findings underscore the promise of learning objectives to advance transformers' capacity for long-term planning.

## 1 Introduction

Transformers trained to predict the next token in a sequence have become the de facto approach in today's best language models (Dubey et al., 2024; Gemma, 2024). Despite their remarkable success, such transformers encounter challenges when tasked with planning and decision-making over extended horizons. This limitation becomes particularly evident in tasks requiring foresight such as maze navigation.

To effectively navigate a maze, a model must have the foresight to plan ahead multiple steps. The de facto next token prediction training approach, however, offers no explicit mechanism to predict multiple steps ahead or revisit the path taken so far. The model is trained to only predict the next step in the input sequence given the previous steps. Prior work has shown next token prediction can fall prey to shortcuts in navigation tasks, particularly as path complexity increases (Bachmann & Nagarajan, 2024). Consequently, we ask: *Can explicitly learning to predict multiple steps ahead (and backwards) improve transformers' ability to navigate mazes?*

To answer this question, we isolate the effect of learning objectives by training transformers from scratch to navigate mazes. Inspired by prior work to remedy shortcomings of next token prediction (Bachmann & Nagarajan, 2024; Gloeckle et al., 2024), we explore the the MLM-$\mathcal{U}$ objective from Kitouni et al. (2024a) as an alternative to next token prediction. MLM-$\mathcal{U}$ proposes masking arbitrary subsets of the input sequence to explicitly predict a variable number of steps ahead and backward as shown in Figure 1. We then assess whether MLM-$\mathcal{U}$ by explicitly predicting multiple-steps during training can improve transformers' performance on maze navigation.

We operate with a collection of mazes with varying levels of grid-size complexities. Two common types of mazes generation approaches are studied that differ in shortest path solution lengths as well

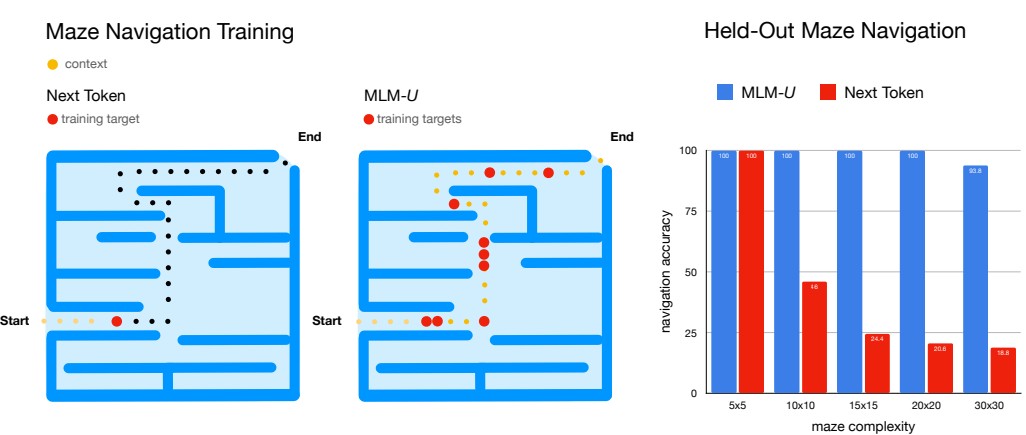

Figure 1: **MLM-$\mathcal{U}$ predicts multiple steps ahead and backward.** Standard autoregressive training only (explicitly) predicts the next step. We compare 8M parameter transformer models trained with autoregressive next token prediction versus MLM-$\mathcal{U}$ training objectives. Maze complexity is defined in terms of the maze grid size.

as maze text representations. For one setting, we train transformer models for both objectives, standard next token prediction and MLM-$\mathcal{U}$. In the other setting, we compare MLM-$\mathcal{U}$ against published results on next token training from Lehnert et al. (2024). Finally, we compare learning objectives across several transformer model sizes by measuring maze navigation, data sample efficiency, as well as training efficiency in terms of GPU hours to convergence.

Our results indicate MLM-$\mathcal{U}$ can improve maze navigation accuracy and training efficiency compared to standard next token prediction. Remarkably, we find a transformer trained with MLM-$\mathcal{U}$ outperforms larger transformers trained with next token prediction using additional supervision from A* search traces (Lehnert et al., 2024). Specifically, relative to standard next token prediction training, we find that:

1. MLM-$\mathcal{U}$ considerably improves transformers' ability to navigate mazes.

    - MLM-$\mathcal{U}$ outperforms comparable next token transformer models across every maze type and grid size complexity tested. For example, an 8M parameter transformer trained with MLM-$\mathcal{U}$ can perfectly solve all mazes of grid sizes up to 20x20, whereas next token training peaks at 20.6% navigation accuracy on held-out 20x20 test mazes (shown in Figure 1).

    - MLM-$\mathcal{U}$ outperforms next token transformers trained with additional A* search trace supervision on complex mazes. For example, on 30x30 mazes an 8M parameter transformer reaches 85.5% navigation accuracy with MLM-$\mathcal{U}$, improving on the 70.2% navigation accuracy of a 175M parameter transformer trained with next token prediction and additional A* search trace supervision.

2. MLM-$\mathcal{U}$ training is 4x more data-efficient in terms of training samples. For simpler mazes (5x5) solved by both MLM-$\mathcal{U}$ and next token prediction, MLM-$\mathcal{U}$ is 2x more efficient in GPU hours needed for convergence.

3. MLM-$\mathcal{U}$ benefits from scaling to larger transformers for more complex mazes. For example scaling MLM-$\mathcal{U}$ from a 3M to an 8M parameter transformer boosts performance from 85% to perfect navigation on 20x20 mazes.

These findings suggest that the learning objective is critical to transformer's maze navigation abilities, offering a promising direction for future research in long-horizon planning tasks.

## 2 RELATED WORK

**Standard next token trained transformers struggle with navigation and planning** Ivanitskiy et al. (2023b) show transformers trained on maze navigation tasks learn internal states that allow a decoding of the entire maze. Despite this emergent state however, Bachmann & Nagarajan (2024) shows the limits of next token prediction objectives for basic graph navigation tasks. In particular, the work identifies a Clever-Hans cheat based on shortcuts in teacher forced training similar to theoretical shortcomings identified in Wang et al. (2024b). This demonstrates that while transformers can represent world states for mazes, they may struggle in planning that requires significant foresight. A remedy found by Bachmann & Nagarajan involves removing the teacher forced supervision. Their view inspired us to look further into the training objective to encourage more explicit planning.

**Deep Learning approaches to maze navigation** Many deep learning approaches for maze navigation use reinforcement-learning objectives (Akmandor et al., 2022; Wang et al., 2024a; Tamar et al., 2016; Wang et al., 2024c; Kong et al., 2024). Liu & Borisyuk (2023) compares the navigation strategies learned by reinforcement learning to those observed in animals suggesting some similarities in learning dynamics. Janner et al. (2022) study reinforcement learning reward modeling with a diffusion objective with applications to planning tasks including maze navigation. While reinforcement learning approaches excel at tasks involving interaction and games, reinforcement learning has played a relatively minor role in foundation model pretraining. Outside of reinforcement learning approaches, Lehnert et al. (2024) successfully train transformers with the next token objective to perform maze navigation. Crucially, they can vastly improve performance via additional supervision. By exposing the model to a trace of an A* algorithm solving the maze, they gain significant performance and data efficiency. Interestingly, just like in Bachmann & Nagarajan (2024), the remedy to failure on a navigation task seems to involve changing the supervision structure. We directly compare this approach with the MLM-$\mathcal{U}$ objective trained without any supervision from A* search traces.

**Diffusion Learning Objectives** Kitouni et al. (2024a) used MLM-$\mathcal{U}$, which can be seen as a diffusion objective (Austin et al., 2021; Kitouni et al., 2024b), to mitigate the reversal curse in language modelling (Berglund et al., 2024), where models trained to answer questions in one way can not generalize to an inverse, semantically equivalent formulation. They also show that MLM-$\mathcal{U}$ performs well in the graph navigation task from Bachmann & Nagarajan (2024). Sahoo et al. (2024); Austin et al. (2021); Li et al. (2022) incorporate diffusion objectives in masked language modeling for general purpose language models. He et al. (2022) adds a diffusion objective to further train a pretrained BERT model showing improvements over standard BERT training in terms of perplexity and BLEU score on language tasks.

## 3 THE ROLE OF LEARNING OBJECTIVES IN MAZE NAVIGATION

We examine how the standard next token learning objective manifests itself in maze navigation, a task requiring planning multiple steps head. We contrast next token prediction with MLM-$\mathcal{U}$, a training objective explicitly encouraging predicting multiple steps ahead and backward.

### 3.1 PREDICTING THE NEXT STEP WITH STANDARD TRAINING

The de facto learning objective used to train language models is next token prediction. This objective, which is also referred to as an autoregressive (AR) or causal-masked prediction objective, when paired with the transformer architecture has shown great success in language tasks at scale. Specifically, given a sequence of inputs $x_1, x_2, x_3, \ldots, x_n$, the next token learning objective minimizes

$$L_{\text{next token}} = -\sum_t \log P_\theta(x_{t+1}|x_{1:t}) \qquad (1)$$

where $t$ indicates the index of the input sequence. This simple objective maximizing the probability of the next token given the previous tokens in the sequence has led to remarkable fluency in language tasks (Dubey et al., 2024; Gemma, 2024). However, transformers trained with next token prediction exhibit limits in terms of planning.

**Standard next token prediction does not seem to encourage explicit multi-step planning.** In maze navigation, as shown in Figure 1, next token prediction amounts to predicting *only the next step* given the path so far. The learning objective in Equation (1) does not explicitly encourage predicting multiple steps ahead. Bachmann & Nagarajan (2024) suggests the lack of multi-step prediction in standard next token training limits transformers' ability to navigate even simple graphs. One pitfall highlighted by Bachmann & Nagarajan (2024) is that models fall prey to short-sighted shortcuts such as the Clever-Hans cheat, show because the model does not plan far enough ahead. Dziri et al. (2024) show similar limits for other multi-step problems, especially as problem complexity increases.

## 3.2 Predicting multiple steps ahead and back with MLM-$\mathcal{U}$

One remedy discovered by Bachmann & Nagarajan (2024) avoids supervision through teacher-forcing by allowing the model to predict the entire path before applying a gradient. However, this approach is slow to train, since it requires the sequential generation steps. Gloeckle et al. (2024) provide an elegant way to reason multiple tokens into the future by having multiple prediction heads. They found this method to have beneficial effects on decoder models of size 13B and above when employing up to 8 prediction heads for the 8 next tokens. Motivated by Gloeckle et al. (2024) we consider an explicit objective predicting multiple tokens both ahead and backwards with a variable, rather than fixed context size. Specifically, we study the MLM-$\mathcal{U}$ objective from Kitouni et al. (2024a) which predicts any subset of tokens given any others as context, hoping to capture long-term context dependence and explicit multi-step prediction.

**MLM-$\mathcal{U}$ explicitly makes predictions multiple steps ahead** MLM-$\mathcal{U}$ proposes masking arbitrary subsets of the input sequence to explicitly encourage the model to predict multiple steps ahead and backwards. The masking ratio, which determines the portion of the input that is masked, is drawn uniformly from [0, 1] thereby encouraging a variable prediction window. Specifically, for a uniformly sampled mask $m_\mu$ with masking rate $\mu$ over the input sequence, the MLM-$\mathcal{U}$ learning objective minimizes

$$L_{\text{MLM-}\mathcal{U}} = - \mathop{\mathbb{E}}_{\mu \in \mathcal{U}} \log P_\theta(m_\mu X | m_\mu^C X) \tag{2}$$

where $m_\mu^C X$ is the context used for prediction, equivalent to the complement of the masked target elements. Incidentally, this method is reminiscent of BERT (Devlin et al., 2019), but with a uniform masking rate and without token substitution. (Kitouni et al., 2024a, see their Figure 2) argue that since the uniform masking rate exposes the model to different length sequences to be completed and to draw information from, there is no distributional shift in a generative inference step.

For maze navigation, as shown in Figure 1, the MLM-$\mathcal{U}$ objective in Equation (2) amounts to predicting multiple steps at various points in the navigation path thereby explicitly planning ahead and back multiple steps.

We study the role of the learning objective in maze navigation by comparing standard next token prediction to MLM-$\mathcal{U}$. We ask: *can modifying only the learning objective to predict multiple steps ahead and back enable transformers to navigate complex mazes?*

## 4 Methods

To study the role of learning objectives for maze navigation, we train transformer models from scratch to generate the shortest navigation path for mazes of increasing complexity. We design our experiments such that transformer models are parameter-matched and trained under identical regimes to isolate the effect of next token versus MLM-$\mathcal{U}$ learning objectives. We assess models' ability to accurately navigate previously unseen mazes as well as their efficiency in terms of training samples and GPU training hours.

### 4.1 Mazes and Their Representations

We consider two maze generation approaches across several levels of grid-size complexities to ensure our findings are not specific to a single type of maze or representation, but hold more generally.

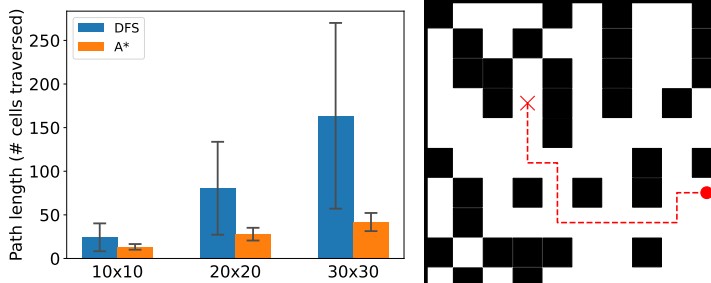

Figure 2: **Left:** Path lengths, measured by number of traversed cells, of A* and DFS mazes for maze sizes 10x10, 20x20 and 30x30 on the validation dataset. Error bars show the standard deviation. **Middle:** Example 10x10 A* maze **Right:** Example 10x10 DFS maze. Both are real randomly selected examples illustrating the difference between encoding walls in cells (A*) versus edges with longer paths (DFS).

**DFS mazes**   First, we utilize the maze generation method from Ivanitskiy et al. (2023a) to generate 2 dimensional mazes via the randomized Depth First Search (DFS) method. This method works by constructing a path from a uniformly random start node in a depth-first manner. This generation approach yields long paths (relative to A* mazes described below), but does not allow ambiguity: the shortest path is also the only path that does not backtrack and thus overlap with itself. An example 10x10 DFS maze in show on the right panel of Figure 2. The mazes are serialized into strings that enumerate the edges of the maze connection graph as a set of tuples. The start node, goal node and solution path are appended to form the full text that the model trains with. We generate 500k mazes across five levels of complexity as measured by the grid size of the maze spanning 5x5, 10x10, 20x20, and 30x30.

**A\* mazes**   Second, we use the deterministic A* maze dataset from Lehnert et al. (2024). Start and goal cell were uniformly sampled in a 2-dimensional grid with walls randomly placed in 30–50% of cells (see middle panel of Figure 2). The shortest paths are discovered via the A* algorithm and added to the dataset if the shortest path is at least of length $L$, where $L$ indicates the maze grid size (for an $LxL$ maze). In A* mazes, grid cells are tokenized with individual tokens for x and y coordinate, which increases the input sequence length relative to the graph tuple encoding used for DFS. In both datasets, the solution path is the last part of the string. In contrast to the DFS mazes, however, A* mazes have many possible solutions, out of more than one are possibly the shortest ones. Lehnert et al. (2024) experiment with both randomly and deterministically (heuristically) choosing the shortest path that the model sees as ground truth. We choose 10x10, 20x20 and 30x30 mazes from the deterministic setting, see Appendix D.2 for additional details.

Together these maze generation approaches allow us to study mazes of varying complexities (in terms of grid size), differing distributions of shortest path lengths, as well as different maze text encoding approaches. In Figure 2 we show the distribution differences between solution path lengths for DFS versus A* mazes across three levels of grid-size complexities. Additionally in the middle and right panels, we show sample generations for DFS and A* mazes.

### 4.2   STANDARD NEXT TOKEN PREDICTION AND A* SEARCH DYNAMIC SUPERVISION

We evaluate the standard next token prediction learning objective for maze navigation. To do so, we train transformers from scratch on text representations of maze solutions similar to Ivanitskiy et al. (2023b). Mirroring the objective of modern language models the transformer predicts the next token based on the previous tokens in the maze solution path (see Equation (1)). We investigate various transformer model sizes to understand the effect of model scale. We also evaluate the standard decoder-only transformer architecture as well as the encoder-decoder architecture from Lehnert et al. (2024). Finally, to better contextualize our findings we also report the next token model from Lehnert et al. (2024) trained with additional A* search trace supervision for the A* maze setting.

Table 1: MLM-$\mathcal{U}$ compared to next token training for 8M parameter transformer-based models trained on 100k maze, solution pairs. We report shortest path accuracy (exact match of all path tokens) for held-out maze of varying complexities based on their grid size. See Table 3 for per token accuracy.

| Maze Navigation (Accuracy) | 5x5 | 10x10 | 15x15 | 20x20 | 30x30 |
|---|---|---|---|---|---|
| Autoregressive | **100** | 45.2 | 24.4 | 20.6 | 18.8 |
| MLM-$\mathcal{U}$ | **100** | **100** | **100** | **100** | **93.8** |

### 4.3 MLM-$\mathcal{U}$

We contrast next token prediction with the MLM-$\mathcal{U}$ objective, explicitly predicting multiple steps both ahead and backward. We closely follow the training setup in Kitouni et al. (2024a), including the encoder-decoder transformer architecture with RoPE positional embeddings (see Appendices D.1 and D.3). Identical to the next token baselines, the MLM-$\mathcal{U}$ objective is trained on text representations of the maze solutions. Generation during inference is done in the same way as for the standard next token baselines, generating one token at a time from left to right, with temperature 0 (argmax). Since the uniform masking rate in MLM-$\mathcal{U}$ (see Equation (2)) exposes the model to different sequence prediction and context lengths, there is no distributional shift in a generative inference step as shown in Figure 2 of Kitouni et al. (2024a). For MLM-$\mathcal{U}$, we also train transformers of varying model scales ranging from 3M to 25M parameters to study the effect of model scale on maze navigation.

### 4.4 EXPERIMENTAL SETUP

To isolate the effect of training objectives, MLM-$\mathcal{U}$ versus next token prediction, we train all models from scratch using an identical setup.

**Training** We train transformers for up to 3000 epochs on 100,000 mazes for each setup. The performance of each model is evaluated on a held-out test set of 2000 mazes with the same configuration as the training set. To ensure the baseline comparisons for next token prediction are competitive, we conduct a sweep over learning rate choices and weight decay values (shown in Appendix B). We select the best choice of hyperparameters based on held-out shortest path accuracy for 10x10 DFS mazes. The architecture used to train MLM-$\mathcal{U}$ is an encoder-decoder (as in Kitouni et al. (2024a), detailed in Appendix D.3), but for next token training in DFS mazes we found a decoder-only architecture to be superior to the MLM-$\mathcal{U}$ encoder-decoder, see Appendix A.2. For A* mazes, we report the best available numbers from Lehnert et al. (2024) for next token prediction.

**Evaluation axes** We evaluate models in terms of maze navigation accuracy, data efficiency as measured by the number of training mazes, and training efficiency in terms of GPU training hours needed for convergence. To assess the correctness of a generated path similar to Lehnert et al. (2024) we compare whether the full path matches the shortest path. We additionally compare the token-wise accuracy in Appendix A.1 to assess navigation paths that only slightly deviate from the shortest path. Finally, to complement the overall maze navigation accuracy, we assess training dynamics by comparing convergence curves on training and held-out tests mazes.

## 5 RESULTS: LEARNING TO NAVIGATE MAZES WITH MLM-$\mathcal{U}$ TRAINING

We compare the next token and MLM-$\mathcal{U}$ objectives via maze navigation accuracy across three dimensions: maze complexity, training data efficiency and computational efficiency. We also investigate scaling laws as well as analyze the training dynamics of MLM-$\mathcal{U}$.

### 5.1 MLM-$\mathcal{U}$ AND STANDARD NEXT TOKEN TRAINING IN DFS MAZES

**MLM-$\mathcal{U}$ outperforms next token prediction for DFS generated mazes.** First, we compare the objectives in the setting with DFS generated mazes described in the first part of Section 4.1. We

train 8M parameter transformer models across mazes with grid sizes ranging from 5x5 to 30x30. We find MLM-$\mathcal{U}$ is able to perfectly navigate mazes of up to a grid size of 20x20 and achieve nearly 3x the performance of next token training on more complex 30x30 mazes as shown in Table 1. For example, even on comparatively small mazes of size 10x10 we find next token performance saturates below 50% accuracy. In contrast, a model of the same size can navigate 30x30 mazes with over 90% accuracy when trained with MLM-$\mathcal{U}$.

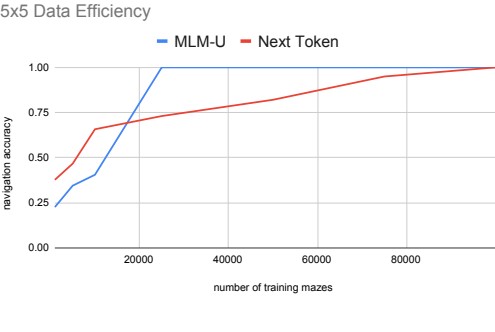

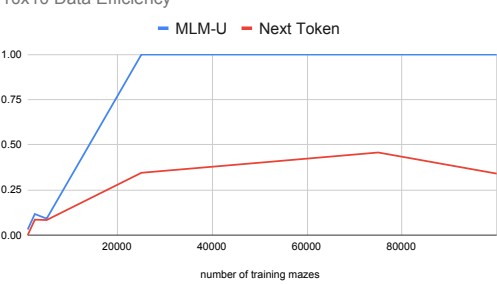

Figure 3: **Training Data Sample Efficiency.** We compare 8M parameter model next token versus MLM-$\mathcal{U}$ held-out accuracy as we vary the number of mazes seen during training. On the left, for 5x5 mazes which both learning objectives can solve, MLM-$\mathcal{U}$ is $4\times$ more data efficient. On the right, for 10x10 mazes we see MLM-$\mathcal{U}$ converges to perfectly solve 10x10 mazes with 25k training samples, where next token performance peaks below 50% accuracy.

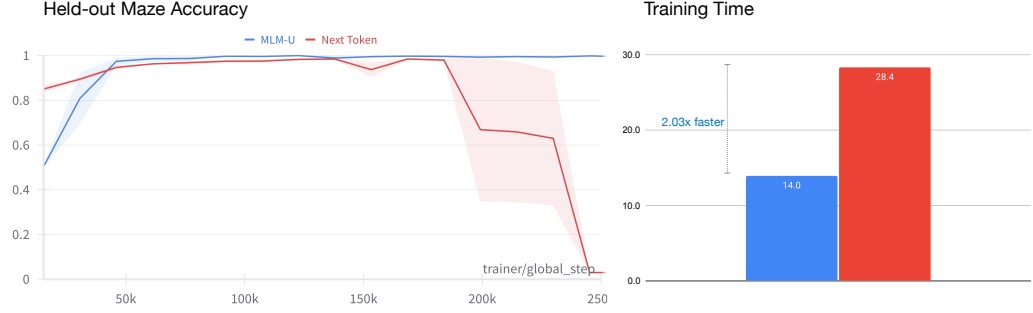

Figure 4: Training efficiency of next token vs. MLM-$\mathcal{U}$ on 5x5 mazes. While both models are able to perfectly solve held-out 5x5 mazes, MLM-$\mathcal{U}$ does so 2.03x more quickly relative to next token. The shaded region shows the standard error across the mean over three random seeds. We also observe overfitting for next token training past 200k training steps whereas MLM-U accuracy remains at near perfect accuracy.On the right, we show the number of GPU hours needed for each training objective to converge.

Table 2: Maze navigation accuracy for MLM-$\mathcal{U}$ training compared to next token training with and without A* search traces for encoder-decoder models trained on 100k A* maze and solution pairs. Baseline numbers are all taken directly from Lehnert et al. (2024). 15M, 175M, and 8M indicate the number of parameters in the transformer architecture used for training. Accuracies refer to an exact match of true and generated path. See Table 4 for per token accuracies in MLM-$\mathcal{U}$.

| Maze Navigation | 10x10 | 20x20 | 30x30 |
|---|---|---|---|
| MLM-$\mathcal{U}$ 8M | **98.5** | **95.2** | **85.5** |
| Next token 15M (Lehnert et al., 2024) | 93.6 | 39.0 | 13.3 |
| Next token 175M (Lehnert et al., 2024) | 94.9 | 53.5 | 19.3 |
| *+ A* trace supervision* | | | |
| Next token 175M (Lehnert et al., 2024) | **98.5** | 90.4 | 70.2 |

**MLM-$\mathcal{U}$ is more data efficient**    To evaluate the data efficiency of MLM-$\mathcal{U}$ relative to that of next token, we train 8M parameter transformer models while varying the number of mazes seen during training. We operate on maze sizes of 5x5 and 10x10 and train both models for 2000 epochs. As shown in Figure 3, we find MLM-$\mathcal{U}$ is able to navigate both 5x5 and 10x10 mazes with only 25k training samples, while next token requires all 100k mazes to reach full accuracy in 5x5 and reaches a peak performance of less than 50% with 75k training samples, suggesting MLM-$\mathcal{U}$ is 4× more data efficient.

**MLM-$\mathcal{U}$ is more computationally efficient on small mazes**    We compare the convergence rates both on training and held-out 5x5 mazes for MLM-$\mathcal{U}$ and next token prediction. We choose this small setting because this is solvable by both objectives. We find as shown in Figure 4 MLM-$\mathcal{U}$ converges 2.17x faster in terms of the number of training epochs. We additionally control for computational overhead in terms of GPU training hours, we find training on the same data for 2k epochs using 8M parameter transformers on 8 Tesla V100 32GB GPUs takes 13.7 hours for next token versus 17.7 hours for MLM-$\mathcal{U}$. Accounting for this additional 7% overhead, we find as shown in Figure 4 **MLM-$\mathcal{U}$ is $\sim 2\times$ more efficient than a comparable next token model on small DFS mazes**. As a caveat, we note that on 10x10 mazes, next token training crosses the 40% performance threshold faster than MLM-$\mathcal{U}$, indicating faster initial learning before saturating at peak of 46% accuracy on held-out test mazes.

## 5.2 MLM-$\mathcal{U}$ and next token training with A* Mazes

**MLM-$\mathcal{U}$ outperforms next token prediction with and without A* search supervision**    In this section, we train models with MLM-$\mathcal{U}$ on the deterministic A* maze dataset from Lehnert et al. (2024) as described in the second part of Section 4.1. We compare those models to the ones trained in Lehnert et al. with and without additional supervision from A* search traces. For example, a nearly 2x larger 15M parameter transformer trained with next token prediction achieves 13.3% navigation accuracy on 30x30 mazes whereas MLM-$\mathcal{U}$ reaches 85.5% navigation accuracy. The results can be found in Table 2. The 8M parameter MLM-$\mathcal{U}$ trained transformer compares favorably with all models from Lehnert et al. trained on 100k mazes. This holds true even when aiding the training with additional supervision provided by the A* search trace, which boosts next token training by a significant margin.

## 5.3 Understanding the training dynamics of MLM-U Compared to next token

**Next token training is more prone to overfit than MLM-$\mathcal{U}$**    We compare the convergence rates both on training and held-out 10x10 DFS mazes for MLM-U compared to next token parameter-matched 8M parameter models in Figure 5. Although we observe faster training convergence for next token models as shown on the left, we see the next token model is not able to generalize from the training data, with performance saturating at around 50%, while MLM-U is able to perfectly solve 10x10 mazes. This suggests while next token training is susceptible to overfitting, where MLM-$\mathcal{U}$ exhibits good generalization without overfitting. We attribute this to the increased difficulty of the

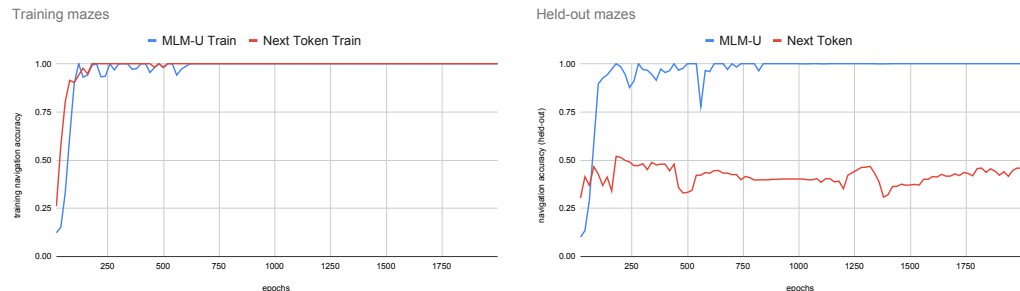

Figure 5: Comparing convergence rates of next token and MLM-U on 10x10 mazes. Left is training accuracy; right is navigation accuracy on held-out mazes.

objective. MLM-$\mathcal{U}$ is tasked to predict any subset of path tokens from any other, while next token training only ever sees the same sequence of conditionals for each maze.

**MLM-$\mathcal{U}$ benefits from scaling to larger transformers for more complex mazes.** Here, we investigate the effect of scaling transformer model size for 20x20 DFS mazes, one the more challenging settings where next token training yields 22% accuracy. As shown in Figure 6 MLM-$\mathcal{U}$ training improves navigation accuracy from 85% to perfect navigation accuracy when transformer model size is scaled from 3M to 8M parameters. For next token prediction, we also observe improvements with transformer model scale, but at a relatively slower rate. A more than 8x increase in model size, from 3M to 25M, for a model trained with the next token objective yields a 43% relative performance improvement.

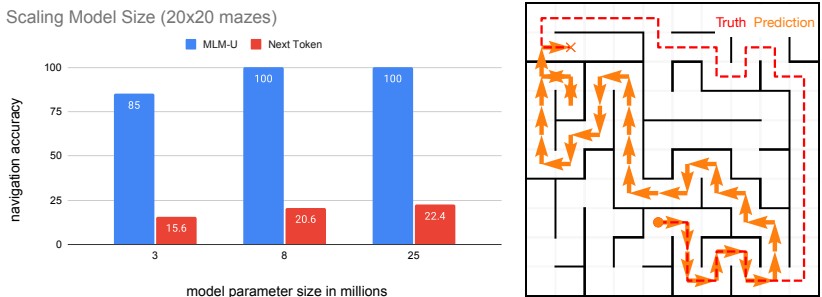

Figure 6: **Left:** Performance of differently sized models (in millions of parameters) across next token and MLM-$\mathcal{U}$ training on 20x20 DFS mazes. **Right:** Example failure of next token training on a 10x10 maze.

## 5.4 POSITIONAL ENCODINGS NEED MORE FLOATING POINT PRECISION

As we scaled MLM-$\mathcal{U}$ training to more complex mazes, we found the precision of the positional encodings to be particularly important for good maze navigation performance. Unlike the learnable ((Radford et al., 2019)) and sinusoidal encodings in the original transformer paper Vaswani et al. (2023) which are added to the input, MLM-$\mathcal{U}$ uses Rotational Positional Encodings (RoPE, (Su et al., 2023)), which bias the query and key vectors in the attention mechanism as a function of their relative positions. To better understand the role of these positional embedding precision we train an 8M parameter transformer MLM-$\mathcal{U}$ on a small set of 100 DFS mazes with increasing grid size complexities. We found with 16-bit precision positional encodings (float 16 via the automatic mixed precision, AMP, package in PyTorch) as shown in Figure 7 (right), MLM-$\mathcal{U}$ generally predicted the correct paths, but failed get the exact positions right, skipping some and duplicating others, resulting in low navigation accuracy on more complex (25x25 and larger) training mazes.

With full 32-bit precision positional encodings however, we found MLM-$\mathcal{U}$ was able to reach perfect navigation accuracy even on these more complex mazes. For example, as shown in Figure 7 on

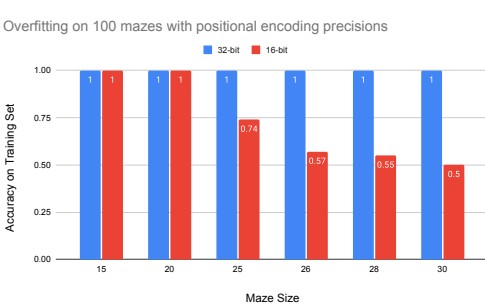

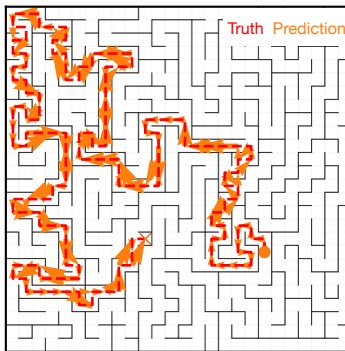

Figure 7: **Left:** Training accuracy of models trained with 16- versus 32-bit positional encoding precision on mazes with different grid sizes. Each model has 8M parameters and is trained on only 100 mazes. For mazes of shape 25x25 and larger, the models cannot overfit on the 100 maze training dataset with only 16-bit positional encoding precision. **Right:** Example 26x26 maze from the train dataset with solution and predicted answer when training with 16-bit positional encoding. The red line presents the true path and the yellow arrows depict the predicted path, generated in a next token left to right fashion. The arrows show inconsistencies and errors on a small scale, but overall follow the correct path.

30x30 mazes MLM-$\mathcal{U}$ only reached 50% navigation accuracy with 16-bit positional encoding precision whereas with 32-bit positional encodings MLM-$\mathcal{U}$ solved 30x30 mazes perfectly. This suggests for larger grid sizes, higher precision in the positional encoding allowed the model to properly map the learned paths to their proper positions on the maze. We observed a similar improvement in performance with larger training data (100k samples) on 30x30 DFS mazes. In particular, by increasing the precision from 16 to 32-bits for positional encodings, MLM-$\mathcal{U}$ performance on 30x30 DFS mazes improved from 40% to 93.8% highlighting the importance of higher positional encoding precision.

While positional encodings have been tailored to next token prediction objectives, less emphasis has been placed on the best positional encoding strategies for masking objectives such as MLM-$\mathcal{U}$. Consequently, the above observations lead us to question whether current approaches are optimal for objectives such as MLM-$\mathcal{U}$. A promising path for training on more complex mazes with larger grid sizes could stem from a better understanding of how best to encode positions for longer-term planning objectives. Therefore, we consider the detailed study of positional bias in masking objectives like MLM-$\mathcal{U}$ crucial for future work.

## 6 DISCUSSION

By adjusting the learning objective from next token prediction to one that explicitly predicts multiple steps ahead and back (MLM-$\mathcal{U}$), we show transformers can learn to effectively navigate mazes. Fortunately, training with an explicit multi-step objective is also more efficient both in terms of training samples as well as GPU training hours and offers nice model scaling benefits with maze complexity. We hope these findings spur the research community to explore learning objectives as a lever to address one of the main limitations of today's best transformer models: multi-step planning. In future work we hope to explore the role of learning objectives in a broader range of multi-step planning tasks.

**Limitations and Future Work**  Of course, such an approach also comes with the typical limitations of transformers, including a fixed context length, which can limit or degrade the training speed of transformers as maze size grows. We observed the importance of positional encodings in MLM-$\mathcal{U}$ training, particularly for more complex mazes. We suggest that there is more understand about the role of positional encodings for planning and identify this as important future work. Furthermore, we acknowledge the increased hardness of the MLM-$\mathcal{U}$ objective. Instead of predicting the same token always with the same context, the context is randomly sampled every time the same training data is observed. For a sufficiently long sequence, the model will never see the same problem twice

due to the exponentially increasing number of possible contexts. We cannot say how this impacts generalization speed in general, although we saw some favorable evidence in this work. In an effort to keep the comparison as straight forward as possible, we used MLM-$\mathcal{U}$ exactly as described in Kitouni et al. (2024a). However, multiple improvements are possible. At inference time, it might be beneficial to generate tokens according to some heuristic about model certainty as opposed to left-to-right. Additionally, the uniform masking rate applied the same way to each token is certainly the simplest, but unlikely the optimal method. A semantic heuristic could favorably impact performance. A possible intuition here is that for many mask realizations, the problem is too easy or too difficult for the model, and it wastes time in those batches. Instead, over-sampling masks that make the problem hard but solvable might yield vastly increased convergence speeds.

In all, these findings shine light on a promising path forward for research to improve long-horizon planning in transformers, with lots of potential for future work.

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

## A  ADDITIONAL RESULTS

### A.1  PER TOKEN RESULTS

To evaluate the possibility of the generated paths deviating only slightly from the shortest paths, we also compute the token-wise accuracy of the generated paths compared to the shortest path. In Table 3 and Table 4 we present per-token accuracies for the experiments from Table 1 and Table 2.

Table 3: MLM-$\mathcal{U}$ compared to next token training for 8M parameter transformer-based models trained on 100k maze, solution pairs. We report per-token shortest path accuracy for held-out maze of varying complexities based on their grid size. Same as Table 1, but including per token accuracies.

| Maze Navigation (Accuracy) | 5x5 | 10x10 | 15x15 | 20x20 | 30x30 |
|---|---|---|---|---|---|
| Autoregressive (per token) | **100** | 46.0 | 32.2 | 25.4 | 25.1 |
| Autoregressive (full path) | **100** | 45.2 | 24.4 | 20.6 | 18.8 |
| MLM-$\mathcal{U}$ (per token) | **100** | **100** | **100** | **100** | 95.8 |
| MLM-$\mathcal{U}$ (full path) | **100** | **100** | **100** | **100** | 93.8 |

Table 4: Maze navigation accuracy for MLM-$\mathcal{U}$ training for encoder-decoder models trained on 100k A* maze and solution pairs, per token and full path accuracies. Refer to Table 2 for baselines.

| Maze Navigation | 10x10 | 20x20 | 30x30 |
|---|---|---|---|
| MLM-$\mathcal{U}$ 8M (full path accuracy) | **98.5** | **95.2** | **85.5** |
| MLM-$\mathcal{U}$ 8M (per token accuracy) | **99.7** | **97.2** | **96.5** |

## A.2 COMPARING TRANSFORMER MODELS FOR NEXT TOKEN TRAINING

We compare two choices of architecture for autoregressive training with transformers: 1) the standard decoder architecture commonly used in modern language models, 2) the encoder-decoder architecture used for MLM-U. We train two 8M parameter transformer models with each of these architectures on 100k DFS 10x10 mazes and evaluate performance on held-out mazes. As shown in Figure 8, we find the common decoder-only architecture converges more quickly and generalizes better than the comparable encoder-decoder architecture. We use the stronger decoder-only baseline for our experiments.

## B ABLATIONS FOR HYPERPARAMETERS

We conduct hyperparameter ablations for learning rates Figure 9 and weight decay in Table 5. We train the next token model with 8M parameters for 500 epochs on 100k 10x10 training mazes and evaluate per-token held-out accuracy to select the best learning rate. Based on this sweep we select 0.001 as the learning rate we use for all our experiments. For MLM-$\mathcal{U}$ we found learning rates to have negligible effect beyond an upper bound to ensure training stability. We select 0.001 as well. We found large weight decay values to be detrimental for next token training, see Table 5. In MLM-$\mathcal{U}$, we generally don't see overfitting and therefore also don't need any weight decay. We choose $10^{-4}$ for next token and no weight decay for MLM-$\mathcal{U}$. We found training to be most stable with the AdamW optimizer with beta values $\beta_1 = 0.9$ and $\beta_2 = 0.999$ and batch sizes of 128 and above.

We evaluate models of two different sizes: 8M parameter models with a width of 128, a depth of 40 and 4 heads per attention layer. For 25M parameter models, the width is 256 with a depth of 32 and

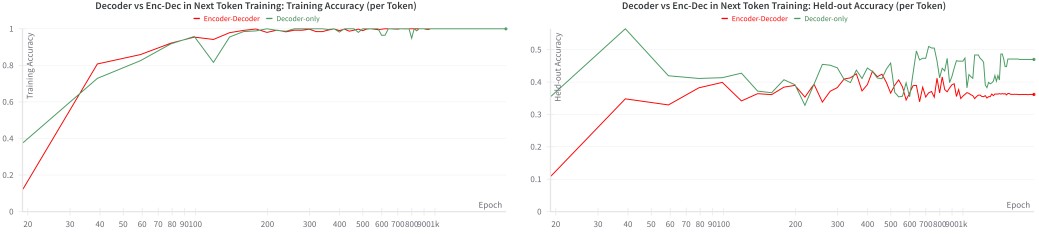

Figure 8: We compare two choices of architecture for next token training with transformers on 10x10 DFS mazes.

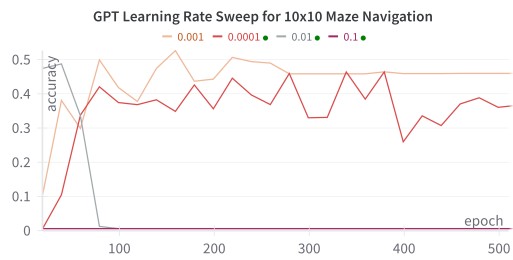

Figure 9: Learning rate ablations for autoregressive (8M parameter) model training on 10x10 mazes for 500 epochs. The y-axis shows the accuracy on held-out 10x10 mazes.

Table 5: Impact of weight decay on GPT training on DFS mazes

| Weight decay | $10^{-2}$ | $10^{-3}$ | $10^{-4}$ | $10^{-5}$ |
|---|---|---|---|---|
| Val Acc (%) | 41.0 | 41.1 | **43.7** | 43.5 |

also 4 heads per attention layer. In the case of an encoder-decoder, both encoder and decoder have depth/2 layers. During development of the experiments, we found that deeper models generally do slightly better in the 8M parameter setting, both innext token training and in MLM-$\mathcal{U}$.

## C  MLM-$\mathcal{U}$ AND NEXT TOKEN FAILURE MODES

In Figure 10 we give some visual examples of MLM-$\mathcal{U}$ failure modes on 30x30 DFS mazes using the 8M model from Section 5.1. Often, the general path taken is mostly correct, but it takes a wrong turn or two and then backtracks to follow the right track, possibly ending up only a few steps short of the goal node. Figure 11 shows example failure cases of the next token model. Often, there is a general tendency towards the right path, but we find frequent backtracks, traversals through walls and often completely wrong end points.

Figure 12 shows failures for the 8M model trained on the A* mazes, from Section 5.2. Note that in two of those failure cases (bottom left and right), the paths predicted are equivalent shortest paths. However, since we are checking for exact match in the deterministic A* setting from Lehnert et al. (2024), those count as faulty. In those instances, the model does not seem to have picked up the way in which symmetry between shortest paths is broken in the deterministic dataset. Note that there also exist other failures that cause parsing errors and can therefore not be depicted. Those make up about half of all failure cases in the validation dataset for this 8M MLM-$\mathcal{U}$ model. The failure cases in Figure 13 for the 30x30 A* maze case are conceptually similar. However, the model fails in some additional ways. For instance, it sometimes misses –or malforms– a step, which ends up being displayed as a diagonal move (left top and bottom). Or it predicts traversal through a wall (top right). The bottom right path is a proper shortest path, but the model does not predict the last move correctly.

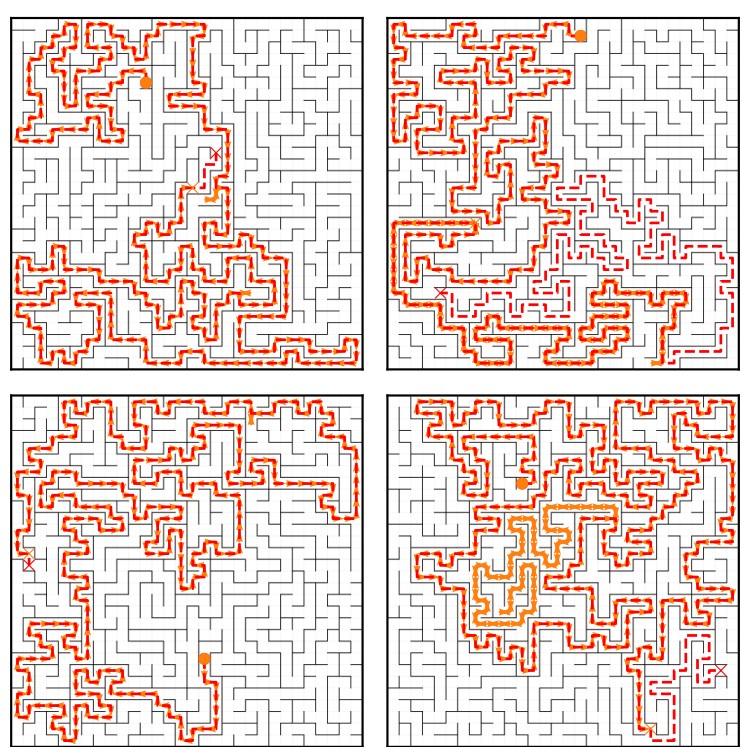

Figure 10: MLM-$\mathcal{U}$ failure examples on 30x30 DFS mazes.

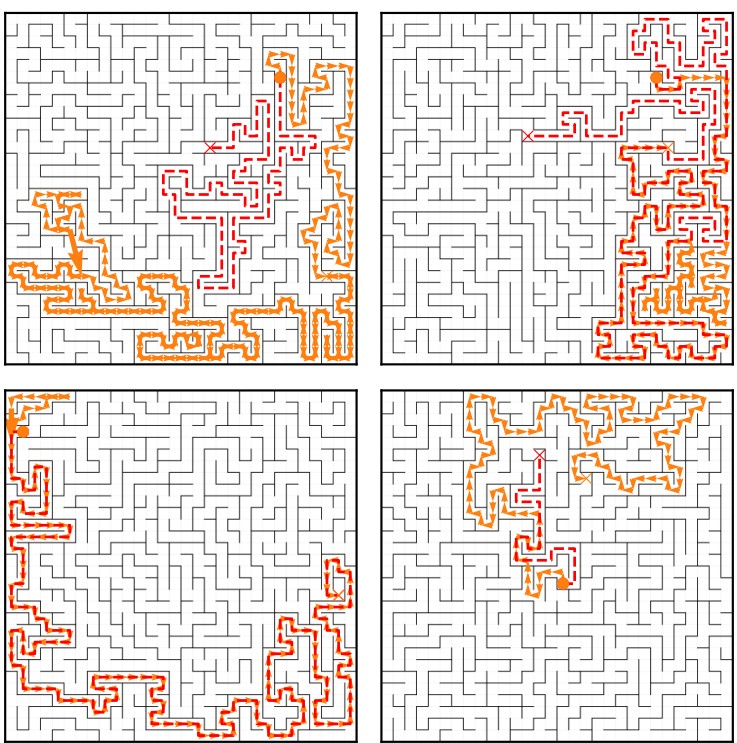

Figure 11: Next token failure examples on 30x30 DFS mazes.

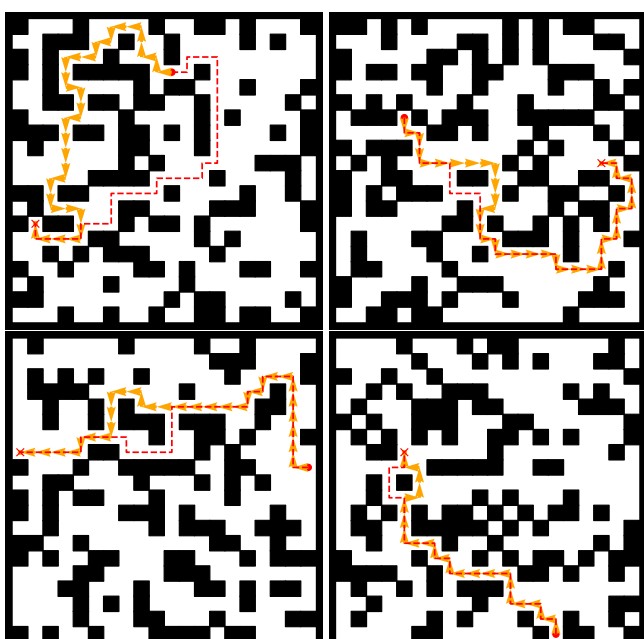

Figure 12: MLM-$\mathcal{U}$ failure examples on 20x20 A* mazes.

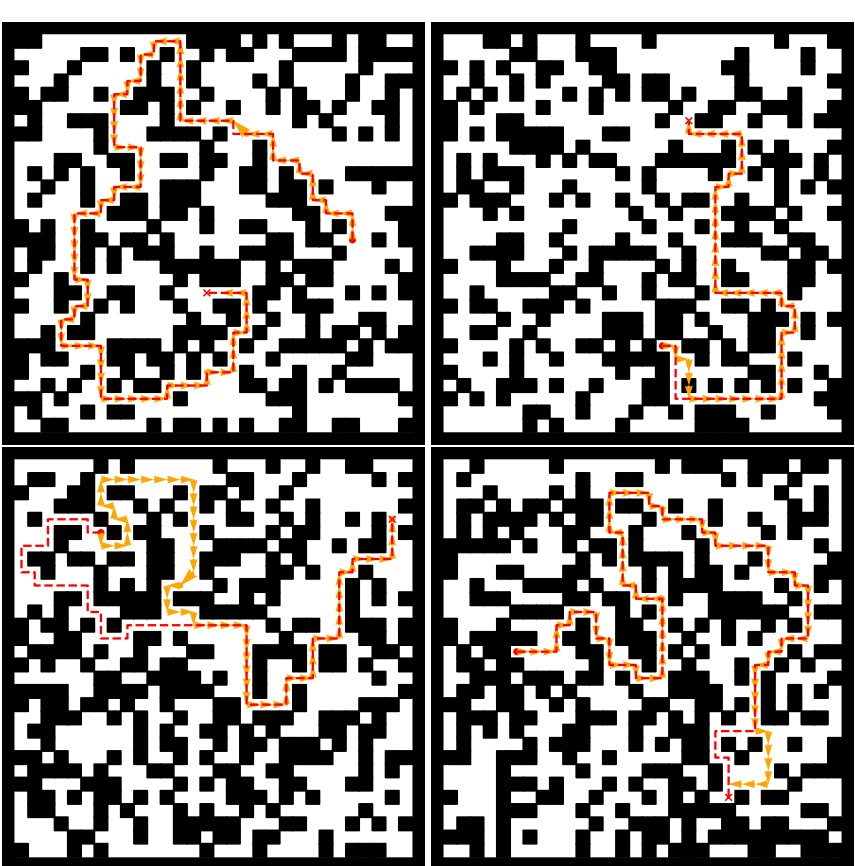

Figure 13: MLM-$\mathcal{U}$ failure examples on 30x30 A* mazes.

# D  MORE DETAILS ON THE EXPERIMENTAL SETUP

## D.1  MLM-$\mathcal{U}$ TRAINING

The MLM-$\mathcal{U}$ models are exposed to the same maze representation, start and end cells and subsequent solution path. Unlike the next token baselines the loss is not a next token prediction loss, but a masking loss reminiscent of the BERT training objective. Tokens are masked with a specific probability and the objective judges the model predictions on the masked tokens via the cross-entropy. In BERT, the masking rate is fixed, but MLM-$\mathcal{U}$ draws masking rates uniformly for each batch. Kitouni et al. (2024a) give an intuition for why uniform masking rates are advantageous. Since the uniform masking rate exposes the model to different length sequences to be completed and to draw information from, there is no distributional shift in a generative inference step, see Figure 2 in Kitouni et al. (2024a).

For this specific case of maze navigation, the only tokens that can be masked are part of the solution path. The model is never tasked to predict the maze representation or start or goal cells. Kitouni et al. (2024a) report that the MLM-$\mathcal{U}$ objective is best trained with a specific encoder-decoder architecture. The encoder has blocks in the layout of GPT-2 with a RoPE positional bias. The decoder input is a sequence of multiple copies of the same learnable token such that the decoder only has information about the positional bias via RoPE. See implementation details in Appendix D.3.

## D.2  MAZE GENERATION DETAILS

We study two different kinds of mazes in this work. They have different properties and are represented in different formats. With that, we aim to demonstrate that our findings are not specific to a single type of maze or representation, but hold more generally.

**DFS mazes**  First, we utilize the maze generation method from Ivanitskiy et al. (2023a) to generate 2 dimensional mazes via the randomized Depth First Search (DFS) method. This method works by visiting all grid cells in a depth-first manner. From a uniformly random start node, it uniformly picks a neighbor cell and removes walls between both cells whenever the target cell was not previously visited. If a cell does not have unvisited neighbors, it is declared a dead end and the algorithm backtracks until a cell with unvisited neighbors is found, starting a new "descent", like in standard depth first tree search. A goal cell is uniformly sampled. This generation algorithm makes for long paths, but does not allow ambiguity. The shortest path is also the only path that does not backtrack from dead ends. The mazes are serialized into strings that enumerate the edges of the maze connection graph as a set of tuples. The start node, goal node and solution path are appended to form the full text that the model trains with. We generate 100'000 mazes for each maze dimension, spanning 5x5 to 30x30.

**A\* mazes**  Second, we use the deterministic A\* maze dataset from Lehnert et al. (2024). Start and goal cell were uniformly sampled in a 2 dimensional grid. Mazes were generated by randomly selecting 30-50% of the cells to be walls and A\* was used to solve those mazes. For an $L$x$L$ maze, the sampled problem is added to the dataset if the solution path is at least of length $L$. In contrast to the DFS mazes, these mazes have many possible solutions, out of more than one are possibly the shortest ones. Lehnert et al. (2024) experiment with both randomly and deterministically (heuristically) choosing the shortest path that the model sees as ground truth. Also unlike the DFS mazes, the text representation describes the set of walls rather than connections and puts the goal and final cell before everything else. In both datasets, the solution path is the last part of the string. Following, the setup in Lehnert et al. (2024) we train on mazes of varying complexities with grid sizes 10x10, 20x20 and 30x30. We train only 100k mazes and reserve 2k mazes each for validation.

**Comparison**  For a direct comparison of the maze setups, refer to Figures 14 and 15. They depict how the prompt and response are made from maze instantiations of the A\* and DFS type.

Notably, the tokenizers for A\* and DFS mazes treat cell representations differently. In DFS mazes each grid cell is one distinct token. This is done to avoid making the sequences too long. In A\* mazes, grid cells are tokenized with individual tokens for x and y coordinate. We believe this

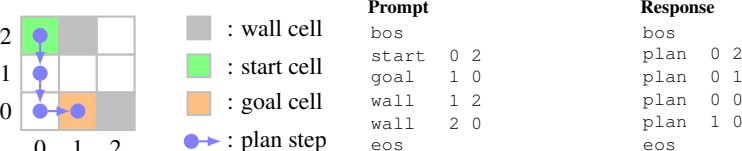

```
                                              Prompt            Response
                                              bos               bos
2                                             start   0 2       plan   0 2
                                              goal    1 0       plan   0 1
1                                             wall    1 2       plan   0 0
                                              wall    2 0       plan   1 0
0                                             eos               eos

   0   1   2
```

Figure 14: A* maze representation, from Lehnert et al. (2024). The maze is serialized as a list of walls and start and goal node. All numbers and words are individual tokens.

```
                                              Prompt                  Response
                                              bos                     bos
2                                             start   (0,2)           (0,2)
                                              goal    (1,0)           (0,1)
                                              (0,0) <-> (0,1)         (0,0)
1                                             (0,1) <-> (0,2)         (1,0)
                                              (0,0) <-> (1,0)         eos
0                                             ...  (all connections)
                                              eos
   0   1   2
```

Figure 15: DFS maze representation. The maze is serialized similarly to the A* setup, but instead of listing walls, connections (i.e. possible movements) are listed, which comes closer to a graph representation with an edge list. Here, each grid cell coordinate (x,y) is a unique token.

presents a better inductive bias than individual tokens for each grid cell, but also increases the sequence length significantly. Since the solution paths are generally much shorter in these mazes, the extra sequence length is affordable. See Figure 2 for a comparison of path lengths between A* and DFS mazes.

### D.3 IMPLEMENTATION OF ENCODER-DECODER

Here we show the exact encoder-decoder algorithm used for MLM-$\mathcal{U}$ training on mazes, as it differs slightly from traditional models. Specifically, the difference lies in the fact that the decoder only sees a sequence of equal embeddings and only gathers information about the mazes from the cross attention with the encoder. Positional information is brought in via RoPE on queries and keys.

---

**Algorithm 1** Encoder-Decoder with MLM-$\mathcal{U}$

---

**Hyperparameters:** $v =$ vocabulary size, $d =$ hidden dim
**Parameters:**
$Enc =$ Stack of Encoder-Transformer blocks (Self attn & RoPE)
$Dec =$ Stack of Decoder-Transformer blocks (Cross attn & RoPE)
$Emb =$ torch.Embedding $[v \times d]$
$p = [1 \times d]$             $\triangleright$ single trainable vector as input to decoder
$Head = Emb^T$ (embedding tied transformer head + softmax)

**Training:**
**Input:** Input sequence $x_{t=1:T}$, Target sequence $y^*_{t=1:T}$ (usually equal to $x$)
**Input:** $m_{\text{pred}}$      $\triangleright$ tokens of interest to calculate the loss over (the solution tokens for mazes)

$m_p \leftarrow$ bernoulli sample a mask over tokens with $p \sim \mathcal{U}(0,1)$      $\triangleright$ MLM-$\mathcal{U}$ here
$m_{\text{pred}} \leftarrow m_{\text{pred}} \cap m_p$     $\triangleright$ these tokens will be predicted (held out tokens of the solution)
$m_{\text{enc}} \leftarrow \neg m_{\text{pred}}$     $\triangleright$ all else: visible context (maze + part of the solution path)
$x_{1:T} \leftarrow Emb(x_{1:T})$
$x_{1:T} \leftarrow Enc(x_{1:T}, \text{attn-mask} = m_{\text{enc}})$
$p_{1:T} \leftarrow \text{expand}(p, [T])$      $\triangleright$ repeat single $p$ to match $x_{1:T}$
$x_{1:T} \leftarrow Dec(p_{1:T}, x_{1:T}, \text{attn-mask} = m_{\text{enc}})$      $\triangleright$ $p_t \rightarrow Q, x_t \rightarrow (K,V)$ in cross attn
$\hat{y}_{1:T} \leftarrow Head(x_{1:T})$
$L \leftarrow \text{CE}(\hat{y}_{1:T}, y^*_{1:T}, \text{mask} = m_{\text{pred}})$      $\triangleright$ Loss, only calculated over $m_{\text{pred}}$

**Inference:** (in AR fashion)
**Input:** Input sequence $x_{t=1:T}$
**Input:** $m_{\text{pred}}$      $\triangleright$ tokens to be predicted
$\hat{y} \leftarrow \text{zeros}[T]$      $\triangleright$ zero tensor of same length as $x$
**for** $T' \in 1:T$ **do**
    **if** $\neg m_{\text{pred}}^{T'}$ **then**      $\triangleright$ $m_{\text{pred}}^i$ is the i'th element of the mask
        $\hat{y}_T' \leftarrow x_T'$      $\triangleright$ don't predict the mazes, only the path
    **else**
        $y_{1:T'} \leftarrow Emb(\hat{y}_{1:T'})$
        $y_{1:T'} \leftarrow Enc(y_{1:T'}, \text{attn-mask} = \neg m_{\text{pred}}^{1:T'})$
        $y_{T'} \leftarrow Dec(p, y_{1:T'}, \text{attn-mask} = \neg m_{\text{pred}}^{1:T'})$      $\triangleright$ $p \rightarrow Q, y \rightarrow (K,V)$ in cross attn
        $\hat{y}_{T'} \leftarrow \text{argmax}(Head(y_{T'}))$      $\triangleright$ AR generation via argmax (Temperature 0)
        $m_{\text{pred}}^{T'} \leftarrow \text{False}$
    **end if**
**end for**
$\hat{y}_{1:T} \leftarrow (\hat{y}_1, ..., \hat{y}_T)$

---

# E  MISCELLANEOUS EXPERIMENTS

## E.1  ORDERED MASKS

One of our motivations for utilizing a training scheme like MLM-$\mathcal{U}$ is that such a scheme enables more explicit reasoning over tokens that are further in the future than the immediate next token, hopefully aiding longer-horizon planning. In light of this view we evaluate the following ablation: In MLM-$\mathcal{U}$ each token in the solution path is masked with some (uniformly drawn) probability, independently of other tokens. Instead, we uniformly pick a position in the solution path and mask all tokens to the right of this position. Then we predict all of those tokens as a function of the context to the left of the chosen position. This method relates closer to the method used to solve the Star-Graph problem in Bachmann & Nagarajan (2024). However, we find that this method is far inferior to MLM-$\mathcal{U}$ in the 10x10 A* maze setting tested. The maximum per-token accuracy observed is 73%, with less than 4% full path accuracy.

## E.2 GENERALIZATION TO SMALLER MAZES

To see whether and how MLM-U and next token trained models perform out of their immediate training distribution, we evaluate models trained on 20x20 DFS mazes on smaller (10x10) mazes. Limitations in length generalization prohibit non-zero accuracies on larger mazes, but experiments on smaller mazes yield interesting results, see Table 6. In all experiments, we tokenize the 10x10 mazes via the 20x20 tokenizer. This is important because the 10x10 and 20x20 tokenizers in our training methods assign different tokens to the grid cells. While next token trained decoders can achieve non-trivial accuracy on smaller mazes out of the box, changing only the tokenizer, MLM-$\mathcal{U}$ can not.

In order to recover good performance in MLM-$\mathcal{U}$, we embed the 10x10 maze into the upper left corner of a random 20x20 maze in an effort to bring the smaller maze closer to the training distribution.

| Configuration | Token Accuracy(%) | Full Path Accuracy(%) |
|---|---|---|
| Next Token | 30 | 21 |
| Next Token embedded in 20x20 maze | 37 | 29 |
| MLM-U | 2 | 0 |
| MLM-U embedded in 20x20 maze | 100 | 100 |

Table 6: Generalization of models trained on 20x20 DFS mazes on 10x10 DFS mazes. Every setting has the 10x10 mazes tokenized via the 20x20 tokenizer. "Embedded in 20x20 maze" means that we put the 10x10 maze into the upper left corner of a 20x20 maze. For all experiments, the 10x10 maze was tokenized via the 20x20 tokenizer.