# OpenReview forum: "Transformers Can Navigate Mazes With Multi-Step Prediction"
_ICLR.cc/2025/Conference — Submitted to ICLR 2025_

### Official Review · Reviewer_vUWK · 2024-11-01

**Soundness:** 2
**Presentation:** 2
**Contribution:** 2
**Rating:** 5
**Confidence:** 4

**Summary:**

The paper "Transformers Can Navigate Mazes with Multi-Step Prediction" investigates whether an alternative training objective, termed MLM-U, can enhance transformers' maze navigation abilities compared to the standard next token prediction objective. The authors note that traditional transformers struggle with long-term planning tasks, such as navigating mazes, as they are typically trained to predict the next token based on the previous tokens. The MLM-U objective, inspired by masked language modeling, aims to explicitly predict multiple steps ahead and backward, encouraging better long-term planning.

The authors train transformers using both the standard next token prediction and MLM-U objectives to navigate various types of mazes, comparing performance across different maze complexities and transformer model sizes. They find that MLM-U significantly outperforms the next token objective in maze navigation, particularly for more complex tasks. Additionally, MLM-U demonstrates improved sample efficiency, reduced training time, and better generalization. For example, transformers trained with MLM-U achieve higher accuracy on mazes up to 30x30 in size and outperform larger transformers trained with next token prediction, even those with additional A* search trace supervision.

Key contributions of the paper include:
1. Demonstration of MLM-U's superior performance in long-term planning tasks, achieving better maze navigation accuracy across all tested maze complexities.
2. MLM-U is found to be more sample-efficient and computationally efficient than next token training.
3. Detailed exploration of the impact of model size, showing that scaling up transformers trained with MLM-U leads to significant gains in performance.

These findings suggest that modifying the learning objective to include explicit multi-step planning can significantly improve the capabilities of transformers in navigation tasks, highlighting a promising direction for further research on long-horizon planning with transformers.

**Strengths:**

### Originality
The paper introduces **MLM-U**, a novel training objective that encourages explicit multi-step prediction both forward and backward, enhancing transformers’ long-term planning capabilities. This approach is original in combining masked prediction with maze navigation, showcasing transformers in a novel, decision-making context.

### Quality
The study is methodologically robust, providing a controlled comparison between MLM-U and traditional next token prediction. It employs diverse maze environments and measures both **accuracy** and **training efficiency**, ensuring the findings are well-supported. The visualizations and detailed experimental analysis add credibility to the work.

### Clarity
The paper is **clearly written**, with effective motivation for MLM-U and well-organized results. Visual aids enhance understanding. Slightly more explicit detail on the training process could further improve accessibility, but overall, the flow is logical and easy to follow.

### Significance
The **MLM-U objective** significantly improves the planning capabilities of transformers, demonstrating better sample efficiency and faster convergence. This work paves the way for **more efficient models** that require fewer computational resources, potentially impacting areas like robotics and game-playing. Its contributions to extending transformers into new sequential decision-making tasks are noteworthy.

**Weaknesses:**

### Limited Novelty in Training Objective
MLM-U's novelty may be limited as similar multi-step prediction methods have been explored in reinforcement learning and trajectory modeling. Including a **comparative analysis** with existing methods would help strengthen the originality.

### Limited Experimental Complexity
The experiments focus only on maze navigation, which may not fully demonstrate MLM-U's broader applicability. Extending experiments to tasks requiring **complex long-term planning** (e.g., robotics or control tasks) would provide stronger evidence of general utility. Testing in **dynamic or uncertain environments** would further demonstrate robustness.

### Limited Baseline Comparisons
The paper mainly compares MLM-U against next token prediction, missing comparisons to stronger baselines like **reinforcement learning approaches** or **graph-based transformers**. Including such baselines would provide a more rigorous benchmark of MLM-U's performance.

### Scalability Concerns
The scalability analysis is limited. More discussion on **scaling challenges** and **computational trade-offs** when using larger models is needed. An ablation study on **hyperparameters** like masking ratios and model depth would help understand how to best tune MLM-U for different settings.

### Insufficient Analysis of Failure Cases
The paper lacks a detailed analysis of **failure cases**. A more in-depth examination of **common failure patterns** and scenarios where MLM-U struggles would provide valuable insights for improving the model and identifying its limitations.

**Questions:**

### Limited Novelty in Training Objective
- Could you provide a more detailed comparison between MLM-U and existing multi-step prediction methods? Specifically, how does MLM-U address limitations in prior approaches?

### Limited Experimental Complexity
- Can you provide results or discuss potential outcomes for applying MLM-U to tasks beyond maze navigation?
- Do you foresee any challenges or necessary modifications when extending MLM-U to dynamic or uncertain environments?

### Limited Baseline Comparisons
- Have you considered comparing MLM-U to reinforcement learning-based methods? What would be the expected advantages or drawbacks?
- Could additional baselines be included in future work to provide a clearer benchmark?

### Scalability Concerns
- Could you expand on the potential challenges when scaling MLM-U to larger models or more complex environments?
- Would it be possible to include an ablation study on hyperparameters like masking ratios to better understand their impact?

### Insufficient Analysis of Failure Cases
- Could you provide more detailed examples of MLM-U's failure cases and potential reasons behind these failures?
- What changes could be made to the model or training process to address these failure modes?

---

> ### Author Response · Authors · 2024-11-21
>
> We’re thrilled to see the reviewer appreciate the robustness of our controlled experiments, originality in exploring the role of learning objectives in maze navigation, and clarity of our presentation.
>
> **Limited Novelty in Training Objective**
>
> Please refer to the general response on the goal of this paper.
>
> We agree similar objectives have been explored in the reinforcement learning, which we acknowledge in our related works. However, reinforcement learning methods have not played a role in the pretraining of foundation models, and our goal is to showcase limitations of Next Token Prediction (NTP). NTP remains the defacto objective for large language model training. Here we show in controlled scientific experiments that NTP is not well-suited to mutli-step navigation tasks. We believe these findings would benefit the research community by providing clear scientific evidence for the advantages of alternative training objectives. Based on your findings, we’ve also added additional wording to make this contribution clear in Section 6 and highlight the strengths of reinforcement learning approaches in the related work in Section 2.
>
>
>
> **Limited Baseline Comparisons**
>
> Thank you for the suggestion to include additional baselines, including approaches from the reinforcement learning literature. While we don’t think these are directly comparable to our approach given the differences in architecture and training setup, specifically in the representation of the mazes.
> However, we do of course acknowledge their superiority (see for instance https://arxiv.org/pdf/2406.08404) and we can see RL as a gold standard to which we hope to rise with general (discrete) foundation models.
>
> **Baselines from prior related work**
>
> Otherwise, in addition to the baselines, we compare MLM-U performance in a rigorous manner against both prior work (Lehnert), including a very challenging baseline that uses extra supervision from A* search traces, and our own experiments with NTP. We consider the more challenging DFS setting which produces longer solution paths, where we perform a sweep over hyperparamaters for the next token baseline and consider multiple architectures (see Appendix A.2).
>
> **Scalability Concerns**
>
> We agree scalability of model size and understanding the role of hyperparameters are important. In Figure 6, we explore model size finding that MLM-U does benefit from scaling model size. When we scale from size 3 to 8M parameters, we find performance improves by 15% to perfectly solve 20x20 mazes. Regarding hyperparameters, we’d like to point out a potential misunderstanding: MLM-U does not have any masking ratio hyperparameters; instead, MLM-U uniformly samples masking ratios per sample with no additional hyperparameters involved. We do perform ablations of the other hyperparameters, in Section 5.3. We also provide an additional experiment in our general response exploring an alternative masking strategy of predicting only in the forward direction, which we show does not work as well as the uniform masking procedure in MLM-U used throughout the paper. We do also highlight several limitations of MLM-U in Section 6, such as the limitation regarding the input context window of transformers.
>
> **Insufficient Analysis of Failure Cases**
>
> Fortunately, MLM-U for mazes up to size X perfectly solves all held-out mazes. In the most complex setting we studied, size 30x30 and 20x20, we did observe failures to solve the mazes in X% of cases. We provide a qualitative analysis of mazes along with their solutions in Appendix C (as shown in Figure 10 and Figure 12). We also describe a failure due to insufficient precision, where we found X. To address some of these challenges, we’d be excited to explore curriculum learning approaches as well as specialized tokenization approaches to better represent world states.
>
> **Other Questions**
>
> > Can you provide results or discuss potential outcomes for applying MLM-U to tasks beyond maze navigation?
>
> We’re excited about the potential for MLM-U to tasks beyond maze navigation. We see maze navigation as a first test bed for demonstrating the capacity of MLM-U to predict multiple steps ahead. We’ve conducted early experiments on graph traversal problems from Stargraph where MLM-U also shows great promise for navigation capabilities. In principle, MLM-U is a general purpose objective that would work with any data that can be turned into a sequence of inputs for a transformer, which is quite a broad set. We would be excited to explore MLM-U in real world planning scenarios. We’ve highlighted this more explicitly in our conclusion section 6

---

> > ### Comment · Reviewer_vUWK · 2024-11-26
> >
> > I thank the authors for the answers.
> >
> > Unfortunately, I do not think that my concerns have been addressed properly. Especially considering that now there is an extension, it would have been possible to apply this loss function on a model based RL work. For instance, you could take STORM: https://arxiv.org/abs/2310.09615 , which is very fast, and change the world model objective function following your approach. Experiments take a couple of hours to be runned. Can you maybe try to apply the loss in three different environments from Atari100k? I would use Freeway, Hero and Boxing. Make sure not to use the trajectories for Freeway, as it would invalidate the meaning of the experiment.
> > Do at least 3 seeds per environment.
> >
> > You should be able to do this on time, experiments do not require a big GPU and are pretty fast. the repo is also quite clean.
> >
> > I'm willing to increase my score, if you manage to report these results.

---

> > > ### Author Response · Authors · 2024-12-02
> > >
> > > We thank you for pointing out the excellent library for STORM, we were able to launch experiments for MsPacMan.
> > >
> > > We explored the baseline model and integrated our encoder-decoder transformer model we use for MLM-U. We want to use this model because we found the GPT-like transformer to not work well with MLM-U in our maze studies. Integrating this model turned out to be involved. With this encoder-decoder model, we could not quite produce the good results that the STORM authors found, no matter if we used MLM-U or next token prediction. This is to be further investigated. I quote here the highest scores that I found during the default sampling procedure for the sampling configuration provided in the `train.sh`
> > >
> > > 1. STORM transformer (gpt-like): 4180
> > > 2. Enc-Dec Next Token: 2260
> > > 3. Enc-Dec MLM-U: 2570
> > >
> > > Keep in mind that the setup proposed in STORM differs considerably in goal and experimental design compared to our work. Specifically, STORM involves multiple models tailored to gaming environments including a CNN-based VAE image processing model, a world simulation model, an actor-critic and 6 different loss terms.
> > > While we think it is possible to get good results, we could not perform the tuning needed to do so. The individual loss curves look very different for each setup, suggesting that there are many dynamics at play that we don't have practical experience with.

---

### Official Review · Reviewer_ZjxN · 2024-11-01

**Soundness:** 2
**Presentation:** 2
**Contribution:** 1
**Rating:** 3
**Confidence:** 3

**Summary:**

This paper explores the applicability of MLM-U (Masked Language Modeling + Uniformly Sampled Masking Ratios) to training transformers to perform simple maze navigation tasks. It finds that compared to next token prediction, MLM-U based transformers can more accurately navigate mazes, completing 93.8% of 30x30 mazes, while next-token prediction models complete only 18.8% (with training on 3K epochs of 100K mazes). Further, MLM-U based transformers are more data and training efficient, saturating navigation accuracy on 5x5 mazes 2x faster than next-token prediction (training efficiency), and with almost 4x fewer training mazes.

**Strengths:**

- The paper tackles an interesting and challenging issue—transformers' limitations in long-term planning.
- The application of MLM-U to the task is well-motivated, and it seems logical and reasonable that using MLM-U-based approaches would improve the performance on maze navigation tasks.
- The paper clearly demonstrates that MLM-U is more efficient as a training procedure than next token prediction given the setup, and the gains reported in the paper are quite impressive.
- The scaling results are impressive, suggesting that this is a promising direction of research for larger models, and more complex tasks.
- As far as I am aware, no existing work has clearly demonstrated the effects of MLM-U, or MLM-based training approaches on maze prediction tasks.
- The paper is generally well-structured, and the visualizations (e.g., maze complexity graphs, accuracy charts) are helpful.

**Weaknesses:**

While the paper does have some notable strengths, there are also several weaknesses:
- Some of the key setup and explanation details are missing from the paper, for example: What is the input representation of the mazes (does it match the Lehnert paper? Or are there modifications)? What is the task definition (how much information is given about the maze - is the full representation given, or only limited information, requiring the model to remember across the trajectory)? What is the definition of "navigation accuracy" (is this the same as A* solution accuracy in the Leonhart paper)? It seems like the paper overlooks some of these basics, without making reference to an identical setup, or clearly explaining in the text/appendix. \
- Given that the setup is identical to the Lehnert paper, I'm not sure that this paper actually contributes much to the conversation on long-horizon planning with transformers. While this is solid evidence that for a toy task such as A* navigation matching, adding MLM-based objectives can help improve the performance of the model, maze navigation itself does not really require foresight (especially in the full-information version of the task), and it's not clear that these results will generalize to other long-horizon planning problems. This paper would be significantly bolstered by adding additional domains (for example, In Lehnert, they solve Sokoban puzzles), or potentially other puzzle setups such as Sudoku, which might have slightly more complex mechanics and still require planning. As it stands, I think that primarily focusing on maze navigation is interesting, but perhaps too limited.
- While the motivation for the paper is that MLM-U allows the model to predict multiple steps both forward and backward, I'm not convinced that the setup actually does this. While the details are a bit thin, it seems like the the MLM approach is applied to full trajectories - which while it does encourage predicting multiple steps in the past, seems to be significantly harder to apply during the inference phase. Indeed, I'm not sure that this encourages "foresight" so much as "hindsight" or "justification", and maybe helps the model with memory instead of planning.
- The paper has no baselines beyond next-token prediction. While the main motivation of the paper is to compare MLM-U with vanilla next token prediction, it would be good to include both the Bachmann & Nagarajan approach (predicting the entire path before gradient) and the Gloeckle approach (multiple future prediction heads) as baselines - given that both of these methods claim to be quite effective for this task, and represent comparable methods.
- There is no statistical analysis of the data, and it seems like many of the results are drawn from a single training run on each of the methods. It would be good to see the variance of the data, particularly in Figures 3 and 4, to understand exactly how variable the convergence is (it seems like Figure 4 in particular could be susceptible to fairly high variance).

**Questions:**

- How is navigation accuracy defined?
- What are the setup details (input representations, task definition, etc.)?
- What precisely is the training algorithm for the approach? Is MLM applied to full trajectories? Or is MLM applied at each next-token prediction step? What are the sets of uniform masks used for MLM-U
- During decoding, how is the length of the initialized X determined? It seems like you need to already know the length of the final trajectory T in order to predict the masked tokens.
- How many of the failure cases are equivalent shortest paths that are not equivalent to the deterministic approach (as discussed in Appendix. C)? Are these a significant portion of the results?
- Are there any explicit failure modes recognized in the MLM-U examples (particularly for larger mazes) - or does MLM-U struggle with any categories of maze structure in particular (i.e. those with higher branching factors)?
- Are there any limits on data efficiency gains with MLM-U? How well does it perform on even larger mazes (perhaps 50x50 mazes)?
- Are there data effects from using a smaller number of mazes (i.e. 100K) over a larger number of epochs (3000) rather than a larger number of mazes/more diverse set of mazes in general? How does this compare if you use more dynamic sets of mazes such as in the Lehnert paper (which uses up to 1M unique mazes)? It would be good to replicate Fig 11/12 from that paper in order to get a better idea of the training data scaling.

Some minor comments:
- There's a couple of un-supported claims in the paper that would benefit from citation (L35: "transformers encounter challenges when tasked with planning and decision-making over extended horizon", L37: "particularly evident in tasks requiring foresight")
- Saying that masking rates are "uniform" is somewhat under-specified, and upon reading the appendix, it seems like "uniform" means that the masking rate is _drawn from a uniform distribution_. It would be good to clarify this use of terminology, and clearly specify the exact distribution used for the masking rate in MLM-U.

---

> ### Author Response · Authors · 2024-11-21
>
> We appreciate your effort in carefully reviewing this paper. We are excited that you consider our experimental results impressive and our writing well-structured. Please also refer to the general response.
>
> **Regarding Weaknesses**
>
> > Some of the key setup and explanation details are missing from the paper, for example: What is the input representation of the mazes (does it match the Lehnert paper? Or are there modifications)? What is the definition of "navigation accuracy" (is this the same as A* solution accuracy in the Leonhart paper)
>
> The setup is the same as the deterministic case in the Lehnert paper. Since it is deterministic, we can employ the "exact match" metric in all experiments on A* mazes. We do describe this exact match criteria in  Appendix C (L718), but we agree that that should be described more specifically in the main text as well. Based on your feedback, we now include an expanded description in the main text in the caption of  Table 2 in the updated paper. Please let us know if you have any further questions regarding the setup. We’d be glad to clarify or update the draft accordingly.
>
> >  I think that primarily focusing on maze navigation is interesting, but perhaps too limited.
>
> We agree maze navigation is of course one setting to study multi-step planning capabilities. We admit that this task does not cover imperfect information cases, but otherwise we consider it a pretty good perfect-information planning proxy.
> We found that proving such a superiority in maze navigation is in itself worth publishing, since the standard next token prediction objective falls short quite significantly. As reviewer ZjXN noted, we also explore the benefits of scaling MLM-U: “The scaling results are impressive, suggesting that this is a promising direction of research for larger models, and more complex tasks,” which we’re eager to further explore in follow-up work.
>
> > MLM-U is applied to full trajectories, which while it does encourage predicting multiple steps in the past, seems to be significantly harder to apply during the inference phase.
>
> We believe this may come down to a misunderstanding of the difference between inference and training for MLM-U.  MLM-U is at least trained to predict tokens further ahead (and backward) of its context, at least with some finite probability. At inference time however, we generate tokens in an autoregressive manner identical to next token prediction as described in Section 4.3.
> We agree there is a question of whether forward and backward token prediction in training is actually what drives the performance is of course unknown to us. We also ran a naive experiment where we mask all future tokens from some point onwards, but found this did not perform as well as multi-token predictions both ahead and back (as described in the general response). We hope this additional experiment as well as the distinction between inference and training for MLM-U have fully answered your question. Please let us know if you have further questions.
>
> > The paper has no baselines beyond next-token prediction
>
> Fair point. Note however that our main goal was to point out the flaw in NTP. We see MLM-U as what one would converge to if one took Gloeckle to its extreme.
>
> > The results are drawn from a single training run on each of the methods. It would be good to see the variance of the data, particularly in Figures 3 and 4.
>
> We agree it’s important to account for the variability across runs, thank you for pointing this out. Based on your feedback we’ve update Figure 4 to show the standard error about the mean for both next token prediction and MLM-U. Fortunately, the results across multiple seeds confirm MLM-U has stable convergence, whereas next token prediction overfits with continued training across the multiple seeds we ran. We hope this addresses your concern regarding variability and provides evidence for the robustness of our findings.

---

> > ### Comment · Reviewer_ZjxN · 2024-11-24
> >
> > Thanks for taking the opportunity to respond to some of the weaknesses.
> >
> > - Key setup details: It is good that this matches the Lehnert paper, however there's nowhere in the paper itself where the setup is explained (for example, what do the prompts look like, or how are the mazes passed to the LLMs). It would be extremely helpful to have this in the appendix somewhere (just as Lehnert does in Figure 1, Section 3.1 and Figure 6).
> >
> > - Limiting experimentation: If the goal of the paper is to scale to multi-step planning, another experiment demonstrating that there is some domain transfer would be necessary. Indeed, to me, the focus on mazes is still a key limitation of this paper, as while it does seem to work in small toy examples in a single domain, the paper could be applied in much more complex scenarios. Also, because this is LLM-focused, there's no guarantee that such planning capabilities exist outside mazes (which may be more in-domain) compared to other planning scenarios. Indeed, in domains such as robotics, LLMs have already been shown to struggle with complex planning [1,2,3,4,5]
> >
> > - Training vs. Inference: This is still unclear to me - if the masking is sampled at training time, and you're performing MLM, this would suggest that the model is always trained in a non-causal way during train time. However, if the model is inferenced using causal modeling, wouldn't such a process be off-policy? If the model is trained in a causal way at training time, wouldn't such a model not be paying attention to future masked states, thus contradicting the claim that foresight is necessary?
> >
> > - Baselines: There are a number of non-LLM baselines that could be considered (i.e. A*, jump-point search for path planning). It makes sense to me to include some of these, especially for maze solving, which can **easily** and **efficiently** be solved by such approaches.
> >
> > - Variance: While it is good that SEM has been added to Figure 4, I would appreciate these numbers in the rest of the paper (i.e. Figure 3, Table 2, Figure 5, Figure 6, etc.).
> >
> >
> > [1] Kambhampati, Subbarao, et al. "LLMs can't plan, but can help planning in LLM-modulo frameworks." arXiv preprint arXiv:2402.01817 (2024).
> > [2] Verma, Mudit, Siddhant Bhambri, and Subbarao Kambhampati. "On the Brittle Foundations of ReAct Prompting for Agentic Large Language Models." arXiv preprint arXiv:2405.13966 (2024).
> > [3] Bohnet, Bernd, et al. "Exploring and Benchmarking the Planning Capabilities of Large Language Models." arXiv preprint arXiv:2406.13094 (2024).
> > [4] Valmeekam, Karthik, et al. "Planbench: An extensible benchmark for evaluating large language models on planning and reasoning about change." Advances in Neural Information Processing Systems 36 (2024).
> > [5] Guan, Lin, et al. "Leveraging pre-trained large language models to construct and utilize world models for model-based task planning." Advances in Neural Information Processing Systems 36 (2023): 79081-79094.

---

> ### Author Response · Authors · 2024-11-21
> **Re Questions**
>
> 1. In this paper, maze accuracy always refers to an exact token match of the entire path. We also provide additional token-accuracies in Appendix to gauge paths that are approximately accurate, but found both measures to provide comparable results. Based on your feedback, we’ve also made this explicitly by updating the caption of Table 2.
>
> 2. Please refer to appendix D.2 and let us know if any details remain unclear.
>
> 3. Please refer to section 4.3 and appendix D.1. We are not sure we understand the question "What are the sets of uniform masks used for MLM-U". To reiterate here, MLM-U works as follows: 1. draw a number m uniformly in [0,1]. 2. Each token in the solution path is masked with probability m.
>
> 4. We decode autoregressively, just like NTP. Therefore we don't need to know the trajectory length. This is not necessarily (or rather likely not) the most effective way of utilizing the model at inference time, but we wanted to keep it the same as in NTP to make this study more controlled. We wanted to make sure that the performance gains come from training, not from inference. See L285ff.
>
> 5. We don't have exact numbers here. About half of the failure modes had parsing errors. The other failures often had consistent paths, but more often than not they were longer than the shortest one. In conclusion, less than 25%.
>
> 6. Good question. The failure modes usually correspond to the longest paths, which might be a result of the autoregressive decoding with finite error probability. Note for instance Figure 10 vs 11, which were randomly picked failures in both NTP and MLM-U. You can see the difference in average path length. We also noticed that the models are very good at keeping paths consistent. It is rare that a model does an illegal step.
>
> 7. It takes a long time to train these models. They are quite deep and the 30x30 maze representations have context lengths of 3k+ tokens, scaling quadratically with maze size. We found ourselves compute limited when attempting to train on 40x40 mazes, for instance.
>
> 8. You are right, this would be interesting, but that takes even longer to train (see 7.) and is therefore deferred to future work. As you see, we instead explored the other direction and checked what happens if you lower the amount of data, see Fig 3.

---

> ### Comment · Reviewer_ZjxN · 2024-11-24
>
> 1. Thanks for the clarification.
> 2. This doesn't actually answer my question - Given that these are sequence models, I would expect at least somewhere in the paper to show the _actual sequences_ passed to the model? How do the prompts work? What are the sampling temperatures? etc. All of these details are quite important, and not discussed at all in the design of the mazes. Similar to Figure 1 in Lehnert et al, these details should be discussed.
> 3. Thank you for clarifying these details.
> 4. See my answer in the other comment - is there not off-policy shift when decoding auto-regressively here?
> 5. Thanks for clarifying, it would be good to report this in the paper.
> 6. Thanks for clarifying, it would be good to report this in the paper.
> 7. Given that compute intensivity is a concern, what is the advantage of using such an approach over a compute efficient approach (such as A*, which can easily and optimally solve any 50x50 maze)? It seems like this should be discussed deeper in the limitations.
> 8. Thanks for clarifying.

---

> ### Author Response · Authors · 2024-11-25
>
> **Key details** We added Figure 14 (from lehnert) and 15 (adapted) to address your concern. As a summary here: The prompts work exactly like in Lehner, without the additional A* trace supervision in the prompts. Specifically, a sequence looks like: <BOS> <maze repr> <start node> <goal node> <insert path here> <EOS>. Sampling temperature is 0 (argmax) (see Enc-Dec description in appendix), given that we are looking for exact matches.
>
> **Limited experiments**
> Thank you for bringing this potential misunderstanding to our attention. Our goal is not to overclaim by suggesting we’ve solved the open-challenge of multi-step planning. We choose maze navigation as an exemplary setting, studied by multiple prior works, requiring multi-step planning where standard next token training still struggles. We are excited about the potential of applying alternative training strategies for other domains. However in this work we found it worthy of note for other researchers in the field, and therefore worthy of publication, to find such a striking difference in performance for next token prediction vs a masking objective for maze navigation.
>
> **Training vs Inference**
> Re policy shift: Just to clarify the procedure: Training is like BERT (so indeed masking), but uniformly sampled masking rate instead of fixed 15%. Inference is exactly like standard AR. In L288ff, we explain that this is not off-policy, referring to Kitouni et al (2024). One intuition that might help: Training like we do is (in expectation) the same objective as training a model in the usual fashion (left to right causal), but instead of always having the mask be causal (tokens are functions of previous tokens), sampling uniformly a permutation of that left to right order to condition tokens instead. This is referred to as permutation language modelling, with XLNet as an example.
>
> **Baselines** We don’t believe methods such as A* are appropriate baselines for this work for two reasons: 1) search algorithms such as A* and DFS are used to generate the original maze solutions used for training. 2) our goal is not to compete with dedicated search algorithms; we make no claims to this effect. Our goal is to isolate the effect of learning objectives for transformers in maze navigation.
>
> **Variance** We’re glad you appreciate our effort to add standard error to Figure 4. While of course including standard errors for as many figures as possible is our ambition, in this case we prioritized based on your feedback doing so for Figure 4. In the 5x5 setting, we found the error bounds for MLM-U < 1% for overall maze navigation path accuracy and observed similarly small margins for other settings. Given this relatively small deviation across runs in contrast to the large margins by which MLM-U outperforms next token training: 100% versus 45.2% (10x10), 100% versus 24.4% (for 15x5) and 100% versus 20.6% (for 20x20), and 93.8% versus 18.8% (for 30x30) as shown in Table 1, the performance differences, as noted by Reviewer vUWK, constitute strong empirical evidence for the benefits of training with MLM-U for maze navigation.

---

> > ### Comment · Reviewer_ZjxN · 2024-11-26
> >
> > Thanks for the additional clarification - while I understand the core of the experimentation process, I think that reviewer f3iu puts it quite succinctly: "this work is mostly about the application of the MLM-U objective, proposed in prior work, to the maze navigation domain [also proposed in prior work]." In light of a lack of larger expansion to multi-step planning (Which I originally considered a key strength of the work), and with a key focus on "[isolating] the effect of learning objectives for transformers in maze navigation," this paper feels incremental at best on a topic which is of limited broader interest.

---

### Official Review · Reviewer_f3iu · 2024-11-03

**Soundness:** 3
**Presentation:** 3
**Contribution:** 2
**Rating:** 5
**Confidence:** 4

**Summary:**

In this work, the authors investigated whether Transformers can perform well in long-term planning tasks, and conduct experiments on the maze navigation task. As the conventional next-token prediction (NTP) objective is notorious about its shortcomings with navigation and planning, the authors propose to leverage MLM-U, an objective that encourages the model to learn with the entire traces in a non-autoregressive manner to better grasp the ability of multi-step planning. In fair experimental settings, the authors have demonstrated that the MLM-U objective outperforms NTP in terms of both performance and efficiency.

**Strengths:**

- A timely work in the field of the planning ability of Transformers. The study with the MLM-U objective is nice to read.
- The superiority of MLM-U over NTP with additional supervision from A* search traces, as well as the efficiency in the use of training data, demonstrates the effectiveness of MLM-U.

**Weaknesses:**

- A general weakness point is that this work is mostly about the application of the MLM-U objective, proposed in prior work, to the maze navigation domain. This limits the novelty of the work, and further demands the empirical experiments to be more sound and sufficient.

For other points, please refer to the Questions listed below.

**Questions:**

- Are all of the experiments covered in this work equipped with the setting that the test data falls in distribution with the training data in terms of the size of the maze? In other words, Did the experiments on 30x30 mazes leverage the traces of 30x30 mazes for training, and further test on 30x30 mazes as well? It would be great to study the generalization of MLM-U in comparison with NTP in terms of the task difficulty. More specifically, two lines of experiments could be added:
  1. Train on traces of N_1 x N_1 mazes, then test on mazes of N_1 x N_1, N_2 x N_2, N_3 x N_3 ... (we can set N_1 < N_2 < N_3, or N_1 > N_2 > N_3; different meanings are implied).
  2. Train on traces of both N_1 x N_1 and N_2 x N_2 mazes, then test on mazes of N_1 x N_1 and N_2 x N_2. This studies whether MLM-U could exploit positive transfer or handle negative interference within data; And compare with NTP to get the understanding of which one is better.

- In Section 5.4, why would the precision of the positional encoding especially matter in this MLM-U training paradigm? Is it related the difficulty of training with the MLM-U objective? More analysis on the failed training dynamics with half-precision positional encodings will further contribute to the soundness of the paper.

- In this work, the Transformer models are trained from scratch using NTP or MLM-U objectives. However, I believe the contributions of the work would be much greater if the authors conduct experiments with LLMs in the finetuning stage: Instead of leveraging the NTP objective in SFT, try using MLM-U and discover whether the most capable open-sourced LLMs nowadays like LLaMa 3.1 could be empowered with the multi-step prediction ability. Then an interesting further question would be whether using MLM-U instead of NTP in the SFT stage would be harmful to other SFT data that is to be trained with the NTP objective. I encourage the authors to conduct additional experiments to respond with at least the first question (the utility of MLM-U in the SFT stage for OSS LLMs like LLaMa) to further improve this work.

I would consider raising my scores if the authors address my concerns properly.

---

> ### Author Response · Authors · 2024-11-21
>
> We’re thrilled the reviewer agrees planning abilities of transformers is a “timely” and important topic. We’re also glad the reviewer found our experimental setting to be fair, providing good evidence for the benefits of alternative learning objectives for better maze navigation.
>
> **Limited novelty**
> While this paper does not propose MLM-U specifically, we believe our contribution is valuable to the research community in providing clear scientific evidence to the benefit of factorization agnostic training for overcoming one of the main limitations of today’s transformers (as you noted as other reviewers): multi-step planning. As reviewer ZjxN noted, “no existing work has clearly demonstrated the effects of MLM-U, or MLM-based training approaches on maze prediction tasks.”
> Our empirical experiments carefully cover two distinct maze solving tasks, across a range of maze grid size complexities and model sizes. We believe these findings would benefit the research community by providing clear scientific evidence for the advantages of alternative training objectives to overcome transformers’ limitations with regard to multi-step planning.
>
> Please also refer to the general statement about the goal of the paper.
>
> **Questions**
>
> > Did the experiments on 30x30 mazes leverage the traces of 30x30 mazes for training [..]?
>
> In our MLM-U baselines, we do not use supervision from any search traces in training. MLM-U is trained only on maze solutions. All models are then evaluated on held-out mazes of the same size seen during training. What we found remarkable is that MLM-U outperformed the baseline from prior work that in addition to the same training data used for MLM-U also uses supervision from A* search traces. MLM-U outperformed this baseline that uses additional supervision by more than 15% for the most complex 30x30 maze setting as shown in Table 2.
>
> > It would be great to study the generalization of MLM-U in comparison with next token in terms of the task difficulty.
>
> In our experiments, following the standard setup from prior work such as Lehnert et al. we train and evaluate all methods on mazes of the same grid size. We agree it would be interesting to explore generalization of models trained on one maze size to solve mazes of other grid sizes. Based on your suggestion we ran additional experiments to this effect. We now include these results in a new Appendix Section E.2 showing that while MLM-U is more sensitive to modifications of the tokenizer, when the maze is embedded in the same tokenization space MLM-U does generalize significantly better than next token to solve mazes of grid sizes other than those used for training.
>
> > In Section 5.4, why would the precision of the positional encoding especially matter in this MLM-U training paradigm?
>
> We hypothesize precision of positional encodings also likely also matters in next token prediction, but to a lesser extent than it does for MLM-U. Intuitively, this is related to the difference training objective: next token training aims at local predictions that do not require predicting long-range steps in maze states. On the other hand, MLM-U leverages higher precision positional encodings as predictions can span the entire range of maze. More explicitly, next token prediction has an additional positional bias via the causal mask, which as shown in prior work (NoPE paper https://arxiv.org/pdf/2404.12224) can remove the need for positional encodings entirely with next token prediction. MLM-U on the other hand—due to its multiple token ahead and backward predictions—has no causal  masking. Instead the positional bias in MLM-U comes from the explicit encoding.
>
> > In this work, the Transformer models are trained from scratch using NTP or MLM-U objectives. However, I believe the contributions of the work would be much greater if the authors conduct experiments with LLMs in the finetuning stage
>
> Thank you for the suggestion! We agree it would be interesting to assess how well MLM-U can serve as a finetuning objective for maze navigation. Our central scientific aim is to assess in a controlled, apples-to-apples comparison the effect of learning objective in maze navigation, an exemplary tasks requiring multi-step planning. In order to do so, we chose to remove as many confounding factors such as the effect pretraining as well as the possible distribution shift in pre-post training arising from finetuning with a different objective than was used in pretraining. Although finetuning falls outside the scope of the scientific claims we target in this work,  se do agree exploring alternatives to next token prediction in post-training is also an interesting and practically valuable contribution given the rise of open-source models.

---

> > ### Comment · Reviewer_f3iu · 2024-11-27
> >
> > Thanks to the authors for the response!
> >
> > For the added Appendix E.2, thanks for the deeper investigation regarding the sensitivity of the tokenizer under different maze settings. However, my concern is that the tokenizer adaptation method described in Appendix E.2 is kinda restricted to the maze problems only, which might not be able to generalize to other planning problems without the explicit spatial accommodation. Do the authors have further ideas regarding this?
> >
> > Nevertheless, I'm raising my score since overall this is an interesting paper, and most of the questions I raised are clarified.

---

### Official Review · Reviewer_C6d6 · 2024-11-04

**Soundness:** 2
**Presentation:** 3
**Contribution:** 2
**Rating:** 3
**Confidence:** 3

**Summary:**

This paper proposes MLM-U, a training objective for improving multi-step prediction for transformers. Specifically, they propose to use MLM-U, an objective which predicts a span based on surrounding information, similar to the early MLM objective in fine-tuning BERT. Differently, they only use masks and don't use token substitution. Though the approach is very simple, they show that fine-tuning the transformer on two synthetic datasets (created with DFS and A*) with MLM-U objective achieves significantly better performance compared with the next-token-prediction loss usually used in decoder-only models. Besides, they find that fine-tuning with MLM-U tends to converge much faster and demonstrates much better generalization capability to unseen mazes. Their findings suggest that MLM-U objective can be effectively used to improve transformer based agents' capability in multi-step prediction.

**Strengths:**

1. The proposed objective MLM-U, though simple, is very effective in improving the performance of multi-step planning for maze navigation.
2. This paper provides a detailed analysis of the convergence speed, and generalization capability of the fine-tuned model, both suggesting that the transformer fine-tuned with MLM-U tends to achieve much better performance.
3. This paper is well-written and easy to follow.

**Weaknesses:**

1. Some implementation details are not clear to me. For example, when using different objectives, is it the case that MLM-U is applied to encoder-decoder architecture while next-token prediction is used for decoder-only architecture? Does the base model architecture influence the final performance (e.g., next token prediction on encoder-decoder architecture)?
2. The paper emphasizes the importance of RoPE positional encoding. However, the experiments related seem to be about training hyperparameters (i.e., fp16 vs. fp32). I feel it's more reasonable to compare with other positional encodings if the author wants to claim the importance of RoPE positional encoding, otherwise it might be better to call it training hyperparameter ablation.
3. The novelty of the MLM-U objective might be small, as it has been widely explored in many encoder-decoder architecture works.

**Questions:**

Please refer to weakness.

---

> ### Author Response · Authors · 2024-11-21
>
> Thank you for your time reviewing our paper. We are happy that you appreciate our detailed analysis of the comparison between next token prediction and MLM-U. We respond to your feedback and specific questions below:
> >Some implementation details are not clear to me.  Does MLM-U use an encoder-decoder architecture while next-token prediction is used for decoder-only architecture?
>
> We’d be happy to clarify the choice of architectures. We found that encoder-decoder architecture works better for MLM-U (and base this choice on prior work). We found decoder-only architecture works better for the next token baseline. In order to give both objectives a fair chance in the competition, we use the best model for each objective. To ensure our results are not the result of the choice of architecture, we ablate the choice of model choice in Appendix A.2. where we explore both encoder-decoder and decoder architecture and choose the stronger baseline for comparisons in the main paper. Please let us know if you have any other questions regarding our experimental setup.
>
> > emphasis on the role of RoPE positional encodings.
>
> Thank you for pointing this out. We agree our presentation could have been better to appropriately emphasize the core message of this section. While we do use RoPE, we do not seek to emphasize this implementation of positional encoding. We see how the title of Section 5.4 is confusing, thank you for making us aware! Based on your feedback, we’ve revised the title of the section in the manuscript to appropriately emphasize the goal of the section. The point we are trying to make in that section is that the positional encoding might have more of an impact in MLM-U like objectives than in NTP, for two reasons: 1. in NTP, one does not strictly need an additional positional bias on top of the causal mask (see the NoPE paper) and 2. since small changes (like the precision used in calculating it) make a big difference (see Fig 7).
>
> > The novelty of the MLM-U objective might be small.
>
> We agree MLM-U isn’t the only objective to explore multi-token prediction masking objectives as detailed in our general response. The value of contribution is in the application of such an objective to the open challenge of multi-step planning in transformers. As noted by reviewer ZjxN, “no existing work has clearly demonstrated the effects of MLM-U, or MLM-based training approaches on maze prediction tasks.” We show via controlled experiments the benefits such an objective can bring in overcoming the multi-step planning limitations of transformers for maze navigation as well as the promise of scaling such an approach.

---

### Author Response · Authors · 2024-11-21
**General Response**

Thanks to all reviewers for the detailed feedback and suggestions.

Based on reviewers’ feedback we’ve revised the manuscript to address several of the reviewers' concerns with changes highlighted in blue.

**We would like to state explicitly the goal of this paper to set expectations:**
The goal is to **demonstrate superior planning capabilities** of transformers when trained with a different objective. We believe this is a timely, valuable contribution to encourage the research community addressing one of the main limitations of today’s best transformer models (as noted by reviewers ZjXN and C6d6): multi-step planning. We show via controlled experiments with identically sized transformers trained on the same data, by only modifying the learning objective, transformers can learn to efficiently navigate mazes.

We choose maze navigation because it is a synthetic task that requires non-trivial planning in the context and has a simple knob to scale complexity via the maze size. **A model has to reason about the context, perform planning of the path by repeatedly referring to this context and then decode/execute** the shortest path. While it is true that this is a full-information environment, the requirements needed to solve the task are otherwise very general. In that respect, more complex problems like Sudoku and Sokoban are interesting and future work, but we believe that the presented evidence is worthy of publication, given that it shows such strong hints for shortcomings of next token prediction.


We appreciate reviewers found our work “timely” (f3iu) in
**Tackling a key limitation of transformers**
-  “an interesting and challenging issue—transformers' limitations in long-term planning.”—ZjxN

**Prosposing a novel application of a well-motivated method:**
- “As far as I am aware, no existing work has clearly demonstrated the effects of MLM-U, or MLM-based training approaches on maze prediction tasks.”—ZjxN
- “The proposed objective MLM-U, though simple, is very effective in improving the performance of multi-step planning for maze navigation.”—C6d6

**Providing clear experimental evidence**
- “The paper clearly demonstrates that MLM-U is more efficient as a training procedure than next token prediction given the setup, and the gains reported in the paper are quite impressive.”—ZjxN
- “This paper provides a detailed analysis of the convergence speed, and generalization capability of the fine-tuned model, both suggesting that the transformer fine-tuned with MLM-U tends to achieve much better performance.”—C6d6
- “Demonstration of MLM-U's superior performance in long-term planning tasks, achieving better maze navigation accuracy across all tested maze complexities.”—vUWK

**Exploring the benefits of scaling this approach**
- “The scaling results are impressive, suggesting that this is a promising direction of research for larger models, and more complex tasks.”—ZjxN
- “Detailed exploration of the impact of model size, showing that scaling up transformers trained with MLM-U leads to significant gains in performance.”—vUWK

---

> ### Author Response · Authors · 2024-11-21
> **Additional Experiments**
>
> In an effort to alleviate some of the concerns about the diversity of experiments, we introduce a couple of additional experiment results that can be found in appendix E and in the main text of the revised manuscript. We summarize them here:
>
> 1. We include a new version of  Fig 4 showing the standard error across the mean for three random seeds as suggested by reviewer ZjxN. The new results suggest MLM-U has stable convergence across multiple seeds, whereas next token prediction overfits with continued training across the multiple seeds we ran.
>
> 2. Another reviewer criticized that we don't provide additional baselines like the solution provided in Bachmann et al (2024), and another reviewer hinted that they think that the benefits of MLM-U in maze solving do not come from increased "foresight", but rather from "justification" and "hindsight". To investigate both concerns, we conduct the following experiment: Instead of training with MLM-U where all tokens are masked independently of each other with some probability, we uniformly pick a position in the solution path and mask all tokens to the right of this position. Then we predict all of those tokens as a function of the context to the left of the chosen position. This relates closely to Bachmann et al, and since the mask is always "oriented forward" and we don't predict tokens "backwards". However, we find that this method is far inferior to MLM-U in the 10x10 A* maze setting that we tested. The maximum per-token accuracy observed is 73%, with less than 4% full path accuracy. This result indicates that future work is needed to understand the failure modes of looking far ahead, but maybe it sheds some light on the concerns of the reviewers.
>
> 3. One reviewer was interested in seeing whether models can generalize beyond the exact maze distribution they were trained on, specifically when changing the maze size. We have no hope of that generalization to larger mazes than the training distribution, because of the problems of transformers in length generalization. We can however achieve non-trivial accuracy when going to smaller mazes. We evaluate 10x10 DFS mazes with models trained on 20x20 mazes. Interestingly, we find that next token training can somewhat generalize to smaller mazes without any changes except adjusting tokenization, whereas MLM-U only performs well when embedding the 10x10 maze into a quadrant of a random 20x20 maze. Without embedding, MLM-U has 0% accuracy and 100% with the embedding trick. Next token training achieves 21% without embedding and 29% with it, see also appendix E.2 in the revision.
>
> We revised the manuscript to clarify our contribution and aspects of our experimental setup.
>
> With these revisions thanks to the thorough feedback from reviewers, we’re confident the updated manuscript would be a valuable contribution to the research community showing the promise moving away from next token prediction in training transformers for improved mutli-step planning in tasks such as maze navigation.
> We remain available to incorporate any further feedback or to answer any remaining questions any reviewers may have.

---

### Meta-Review · Area_Chair_tHUb · 2024-12-23

**Metareview:**

The paper addresses a limitation of transformers trained for next-token prediction (NTP) in planning tasks that involve predicting multiple steps ahead. As a use case, it considers maze navigation. The authors propose an adaptation of MLM-U (Masked Language Modeling with a uniform sampling ratio) for this navigation task. Through experiments on various maze settings, they demonstrate that their approach outperforms the next-token prediction criterion, converges faster, and generalizes better to unseen mazes.

The reviewers appreciate the paper for its clarity and well-motivated approach. They consider it a timely contribution that implements a simple but effective idea, leading to significant improvements compared to NTP training.
However, the reviewers highlight several weaknesses, including limited novelty, restricted applicability that does not extend to other planning problems, and insufficient comparison with baselines. The authors are encouraged to further extend the application of their approach to a broader class of problems.

**Additional Comments On Reviewer Discussion:**

Most reviewers highlighted the limited novelty and lack of generalizability of the approach. Even with the addition of some experiments, their opinion remained unchanged.

---

### Decision · Program_Chairs · 2025-01-22

Reject